# BDNF-TrkB signaling in oxytocin neurons contributes to maternal behavior

Kristen R Maynard[1†], John W Hobbs[1], BaDoi N Phan[1], Amolika Gupta[1], Sumita Rajpurohit[1], Courtney Williams[1], Anandita Rajpurohit[1], Joo Heon Shin[1], Andrew E Jaffe[1,2,3,4], Keri Martinowich[1,4,5]*

[1]Lieber Institute for Brain Development, Johns Hopkins Medical Campus, Baltimore, United States; [2]Department of Mental Health, Johns Hopkins University, Baltimore, United States; [3]Department of Biostatistics, Johns Hopkins Bloomberg School of Public Health, Baltimore, United States; [4]Department of Psychiatry & Behavioral Sciences, Johns Hopkins University School of Medicine, Baltimore, United States; [5]Department of Neuroscience, Johns Hopkins University School of Medicine, Baltimore, United States

**Abstract** Brain-derived neurotrophic factor (*Bdnf*) transcription is controlled by several promoters, which drive expression of multiple transcripts encoding an identical protein. We previously reported that BDNF derived from promoters I and II is highly expressed in hypothalamus and is critical for regulating aggression in male mice. Here we report that BDNF loss from these promoters causes reduced sexual receptivity and impaired maternal care in female mice, which is concomitant with decreased oxytocin (*Oxt)* expression during development. We identify a novel link between BDNF signaling, oxytocin, and maternal behavior by demonstrating that ablation of TrkB selectively in OXT neurons partially recapitulates maternal care impairments observed in BDNF-deficient females. Using translating ribosome affinity purification and RNA-sequencing we define a molecular profile for OXT neurons and delineate how BDNF signaling impacts gene pathways critical for structural and functional plasticity. Our findings highlight BDNF as a modulator of sexually-dimorphic hypothalamic circuits that govern female-typical behaviors.
DOI: https://doi.org/10.7554/eLife.33676.001

*For correspondence:
keri.martinowich@libd.org

†These authors contributed equally to this work

Competing interests: The authors declare that no competing interests exist.

## Introduction

Brain-derived neurotrophic factor (BDNF) is an activity-dependent neurotrophin that binds the receptor tropomyosin receptor kinase B (TrkB) to mediate many aspects of brain plasticity (*Andero et al., 2014*; *Chao et al., 2006*; *Lu, 2003*). Early social experience, especially maternal care, strongly modulates BDNF levels in rodents and *BDNF* methylation in humans (*Branchi et al., 2013*; *Liu et al., 2000*; *Suzuki et al., 2011*; *Unternaehrer et al., 2015*). A unique aspect of BDNF regulation is transcription by nine distinct promoters that generate ~22 transcripts encoding an identical BDNF protein (*Figure 1a*) (*Aid et al., 2007*; *Pruunsild et al., 2007*; *Timmusk et al., 1993*; *West et al., 2014*). Alternative *Bdnf* promoters allow for tight temporal, spatial, and stimulus-specific BDNF expression, which is critical for modulating plasticity in specific neural circuits (*Baj et al., 2012*; *Baj et al., 2011*; *Pattabiraman et al., 2005*; *Timmusk et al., 1994*). We have shown that *Bdnf* promoters I and II significantly contribute to BDNF expression in the hypothalamus and that selective disruption of BDNF expression from these promoters, but not others, causes enhanced aggression and elevated mounting in males (*Maynard et al., 2016*). While it is established that BDNF modulates social behavior in males (*Chan et al., 2006*; *Ito et al., 2011*; *Lyons et al., 1999*), no studies to-date have investigated the role of BDNF-TrkB signaling in influencing female-typical social behaviors, particularly mating and maternal care. This is especially surprising given the wealth of literature

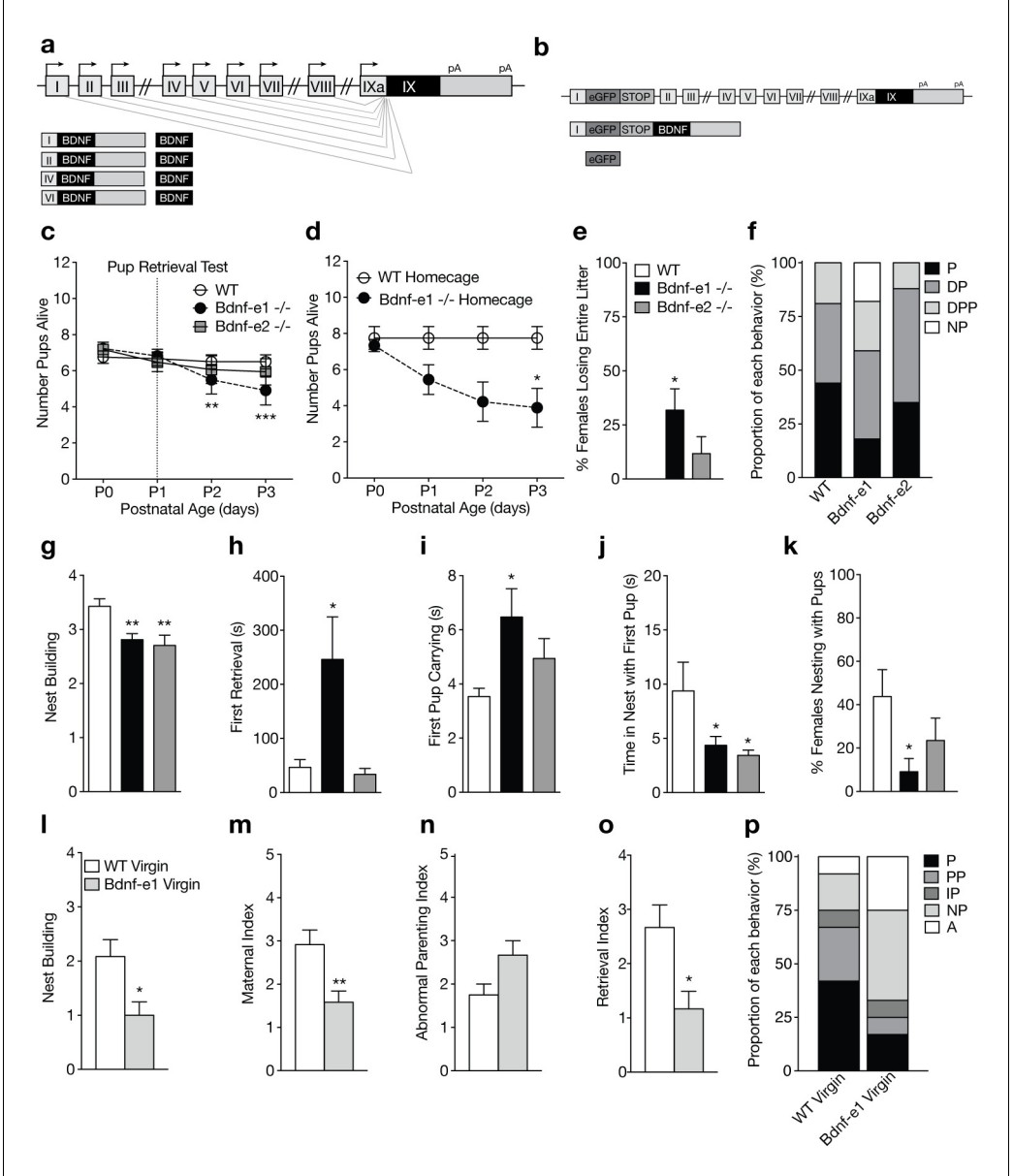

**Figure 1.** Disruption of BDNF from promoters I and II leads to impaired maternal care. (**a**) Schematic of *Bdnf* transcript production. Transcription is initiated from promoters upstream of individual 5'-untranslated regions (UTRs) and spliced to the common coding exon IX. Each transcript produces an identical BDNF protein. (**b**) Design of Bdnf -/- mice using Bdnf-e1 -/- as a representative example. Targeting vectors were designed to insert an enhanced green fluorescent protein (eGFP) upstream of the exon's splice donor site. Bdnf-e1 -/- mice express a *Bdnf*-I-eGFP-STOP-*Bdnf* IX transcript leading to the production of GFP in lieu of BDNF. (**c**) Average litter size over time for WT, Bdnf-e1 -/-, and Bdnf-e2 -/- postpartum mothers exposed to the pup retrieval test one day after giving birth. Bdnf-e1 -/- mothers show significant pup loss compared to WT mothers (2-way ANOVA with mixed effect model; p<0.001). (**d**) Average litter size over time for WT and Bdnf-e1 -/- postpartum mothers remaining in the homecage. Bdnf-e1 -/- mothers show significant pup loss compared to WT mothers even in a naturalistic setting (2-way ANOVA with mixed effect model; p<0.001). (**e**) Percentage of postpartum mothers losing their entire litter by postnatal day 3 (P3). Approximately 1/3 of postpartum Bdnf-e1 -/- mothers lose their entire litter by P3 (1-way ANOVA with Bonferroni's multiple comparisons; p<0.05). (**f**) Proportion of WT, Bdnf-e1 -/- and Bdnf-e2 -/- with different maternal types including parenting (P), disorganized parenting (DP), partial parenting (PP), disorganized partial parenting (DPP), and non-parenting (NP). There is a significant decrease in the proportion of Bdnf-e1 -/- postpartum mothers with parenting behavior compared to WT (one-tailed Mann-Whitney-Wilcoxon rank sum test, p<0.05). Bdnf-e1 -/- postpartum mothers show corresponding elevations in non-parenting

*Figure 1 continued on next page*

*Figure 1 continued*

behaviors. (**g**) Bar graph depicting nest building behavior before parturition. Bdnf-e1 and -e2 -/- show impaired nest building compared to WT (Kruskal-Wallis test with Dunn's multiple comparisons, p<0.01). (**h–j**) Latency to first retrieval (**h**) time carrying first pup (**i**) and time nesting with first pup (**j**) during the pup retrieval test. Bdnf-e1 -/- postpartum mothers show increased latency to retrieval, longer time carrying the first pup to the nest, and reduced nesting time with first pup (1-way ANOVA with Bonferroni's multiple comparisons, p<0.05). Bdnf-e2 -/- postpartum mothers show similar, albeit milder phenotypes. (**k**) Percentage of postpartum mothers successfully retrieving all pups and nesting for at least 2 continuous minutes. Significantly fewer Bdnf-e1 -/- postpartum mothers successfully retrieve pups and continuously nest compared to WT (1-way ANOVA with Bonferroni's multiple comparisons, p<0.05). (**l**) Bar graph depicting nest building behavior of WT and Bdnf-e1virgins 24 hr prior to pup retrieval test. Bdnf-e1 -/- virgins show impairments in nest building compared to WT virgins (Mann-Whitney test, p<0.05). (**m–o**) Maternal (**m**), abnormal parenting (**n**) and retrieval (**o**) indices for WT and Bdnf-e1 -/- virgins during foreign pup retrieval test. Bdnf-e1 -/- virgin females show reductions in maternal and retrieval indices and a strong trend for increased abnormal parenting (Mann-Whitney tests, p<0.01, p<0.05, and p=0.0529, respectively). (**p**) Proportion of WT and Bdnf-e1 -/- virgin females with different maternal types including parenting (**P**) partial parenting (PP), irregular parenting (IP), non-parenting (NP), and attack (A). There is a significant decrease in the proportion of Bdnf-e1 -/- virgins with parenting behavior compared to WT. Bdnf-e1 -/- virgins show corresponding elevations in attack behavior (one-tailed Mann-Whitney-Wilcoxon rank sum test, p<0.05). Data are means ± SEM. (n = 16 WT postpartum mothers; n = 22 Bdnf-e1 -/- postpartum mothers; n = 17 Bdnf-e2 -/- postpartum mothers; n = 12 WT virgins; n = 12 Bdnf-e1 -/- virgins; *p<0.05, **p<0.01, and p<0.001, #p<0.0001).

DOI: https://doi.org/10.7554/eLife.33676.002

The following figure supplements are available for figure 1:

**Figure supplement 1.** Bdnf-e1 -/- females show abnormal mating behaviors.

DOI: https://doi.org/10.7554/eLife.33676.003

**Figure supplement 2.** Bdnf-e1 -/- females have a normal estrous cycle.

DOI: https://doi.org/10.7554/eLife.33676.004

supporting the converse relationship that reduced maternal care impairs BDNF signaling in offspring (*Branchi et al., 2013*; *Liu et al., 2000*; *Unternaehrer et al., 2015*).

BDNF and its receptor TrkB are highly expressed in the paraventricular nucleus (PVN) of the hypothalamus and influence the survival and modulation of magnocellular and parvocellular neurons releasing oxytocin (OXT) (*Castren et al., 1995*; *Kusano et al., 1999*; *Moreno et al., 2011*), a neuropeptide crucial for social behaviors such as pair bonding, mating, and parenting (*Cho et al., 1999*; *Dölen, 2015*; *Marlin et al., 2015*; *Neumann, 2008*; *Williams et al., 1992*). Oxytocin neurons show significant changes in morphology, electrophysiology, and synaptic plasticity during female-typical social behaviors such as parturition and lactation (*Stern and Armstrong, 1998*; *Theodosis, 2002*), but the mechanisms mediating this activity-induced structural and functional plasticity—and whether they extend to other maternal behaviors—are not completely understood. Furthermore, while genes co-expressed with *Oxt* transcripts have been identified (*Romanov et al., 2017*; *Yamashita et al., 2002*), a complete molecular signature of OXT neurons in sexually mature females has not yet been defined. Given that BDNF is a robust modulator of gene expression and has been associated with remodeling of $GABA_A$ receptors in hypothalamic neuroendocrine cells (*Choe et al., 2015*; *Hewitt and Bains, 2006*), we investigated the role of BDNF in regulating gene transcription and plasticity in OXT neurons during female-typical social behaviors. Here we delineate how disruption of BDNF-TrkB signaling in female mice impacts maternal care and OXT neuron gene expression. We define the molecular identify of OXT neurons and demonstrate novel roles for BDNF signaling in the modulation of female-typical social behaviors and OXT neuron plasticity.

## Results

### BDNF derived from promoters I and II regulates maternal care and mating in females

To assess whether Bdnf-e1 or -e2 -/- postpartum mothers show impairments in parenting behaviors, we first examined pup survival from parturition until postnatal day 3 (P3). There was a significant decrease in the number of surviving pups for Bdnf-e1 -/- postpartum mothers compared to WT

(*Figure 1c*; p=0.00023; multinomial regression). Survival analysis using Cox regression showed that pups with Bdnf-e1 and -e2 -/- mothers had a 2.28 and 1.63 times higher probability of death in the first 3 days of life, respectively (p=1.1e-05 and p=0.0028). To rule out that decreased pup survival in Bdnf-e1 -/- was due to maternal stress from pup retrieval testing at P1, we examined pup survival in the homecage. In this naturalistic setting, WT and Bdnf-e1 -/- postpartum mothers were not disturbed following parturition and yet still showed a significant loss of pups compared to WT (*Figure 1d*; p=7.53e-06; log rank test for equal survival). In fact, almost one-third of Bdnf-e1 -/- postpartum mothers lost their entire litter by P3 (*Figure 1e*; $F_{2,52}$= 4.14, p=0.0215). Importantly, pup loss was not due to pup genotype as Bdnf-e1 ±mothers show normal pup survival and Bdnf-e1 and -e2 alleles are not associated with increased lethality (data not shown).

To directly evaluate maternal care in Bdnf-e1 and -e2 -/- postpartum mothers, we used a classic pup retrieval paradigm (*Carlier et al., 1982*) and scored maternal style as well as stereotyped parental behaviors. There was a significant decrease in the proportion of Bdnf-e1 -/- postpartum mothers showing parenting behavior (retrieving and nesting with pups) compared to controls (*Figure 1f*; Mann-Whitney-Wilcoxon rank sum test, W = 111.5, Bonferroni corrected p=0.0466). This decrease was accounted for by an increase in the proportion of Bdnf-e1 -/- postpartum mothers that failed to retrieve or interact with pups (non-parenters). In addition to global parental style, mothers were scored on a number of standard behaviors including nest building (*Figure 1g*; Kruskal-Wallis statistic = 12.95, p=0.0015), latency to first retrieval (*Figure 1h*; $F_{2,50}$= 4.668, p=0.0138), time spent carrying first pup (*Figure 1i*; $F_{2,46}$= 3.414, p=0.0415), and time spent nesting with first pup (*Figure 1j*; $F_{2,51}$= 4.167, p=0.0211). Bonferroni comparisons showed that Bdnf-e1 -/- postpartum mothers were significantly impaired in all these categories compared to WT, while Bdnf-e2 -/- postpartum mothers had milder phenotypes with selective deficiencies in nest building and spending time in the nest with pups. Consistent with a non-parenting style for Bdnf-e1 -/- females, there was a significant decrease in the percentage of postpartum Bdnf-e1 -/- mothers retrieving all pups and nesting for more than 2 continuous minutes (*Figure 1k*; $F_{2,52}$= 3.483, p=0.0380).

Given that postpartum Bdnf-e1 -/- mothers showed striking impairments in maternal care, we also tested Bdnf-e1 -/- virgin females for deficits in pup interaction. We conducted a modified version of the pup retrieval test in which virgin females were single-housed for at least 16 hr and exposed to three foreign pups in the homecage. Like Bdnf-e1 -/- postpartum mothers, Bdnf-e1 -/- virgins showed significant impairments in nest building compared to WT (*Figure 1l*; Mann Whitney, U = 33, p=0.0220). Bdnf-e1 -/- virgins also showed a reduced maternal index (*Figure 1m*; Mann Whitney, U = 28.5, p=0.0092) and a strong trend for an elevated abnormal parenting index (*Figure 1n*; Mann Whitney, U = 39, p=0.0529) compared to WT virgins, which indicates fewer nurturing behaviors such as licking, carrying, and crouching and more harmful behaviors such as kicking, biting, and neglect. Furthermore, Bdnf-e1 -/- virgins displayed a reduced retrieval index compared to WT (*Figure 1o*; Mann Whitney, U = 31, p=0.0144) indicative of a failure to bring pups to the existing nest or generate a new nest around foreign pups. Finally, a significant proportion of Bdnf-e1 -/- virgins showed decreased parenting compared to WT virgins (*Figure 1p*; Mann-Whitney-Wilcoxon rank sum test, W = 39.5, Bonferroni corrected p=0.0285). This decrease in parenting behavior was accompanied by a substantial increase in non-parenting (avoiding pups) as well as biting attacks. Interestingly, while Bdnf-e1 -/- postpartum mothers never showed aggression towards their biological offspring, Bdnf-e1 -/- virgins often attacked foreign pups (*Figure 1p*).

Not only did Bdnf-e1 -/- females show impairments in maternal care, but they also showed perturbations in copulation behavior suggesting reduced mating receptivity (*Figure 1—figure supplement 1*). It was difficult to impregnate pregnant Bdnf-e1 -/- females as male studs paired with these animals frequently showed a variety of genitalia injuries not commonly observed in WT breeder cages (*Figure 1—figure supplement 1a-d*). In particular, there was a significant increase in the percentage of WT male breeders with injured genitalia when paired with heterozygous or homozygous Bdnf-e1 females compared to WT females (*Figure 1—figure supplement 1e*). To test whether male genitalia injuries were due to altered sexual behavior sequences or reduced female receptivity, we paired WT or Bdnf-e1 -/- estrous females with CD1 males and observed mating behavior. While WT and Bdnf-e1-/- estrous females spent equal time exploring the cage (*Figure 1—figure supplement 1f*), Bdnf-e1 -/- females were chased or cornered significantly more than WT females (*Figure 1—figure supplement 1g*; Student's t-test, t = 2.147, df = 22, p=0.0431). Bdnf-e1 -/- females also showed a significant increase in the number of mounting rejections compared to WT females (*Figure 1—figure*

*supplement 1h*; Student's t-test, t = 2.932, df = 22, p=0.0077). Changes in mating behavior were not due to altered hormonal cycling as Bdnf-e1 -/- females entered estrous, as determined by cytological evaluation (*McLean et al., 2012*), approximately every fifth day and the estrous stage lasted for at least 24 hr (*Figure 1—figure supplement 2*). Furthermore, there were no instances of females attacking male genitalia (data not shown), suggesting that male injuries were likely due to altered copulation patterns and failure of females to enter lordosis. These results demonstrate that Bdnf-e1 -/- females have significant impairments in mating behavior associated with subsequent deficits in maternal care.

## Disruption of BDNF-TrkB signaling in OXT neurons leads to reduced maternal care

Bdnf-e1 and -e2 -/- mice show significant loss of BDNF protein in the hypothalamus (*Maynard et al., 2016*), a region containing key neuronal populations, such as oxytocin (OXT) neurons, critical for regulating social behaviors. Given the well-established role of OXT in regulating maternal behavior, we examined oxytocin (*Oxt*) and oxytocin receptor (*Oxtr*) transcript levels in Bdnf-e1, -e2, -e4, and -e6 -/-mice (*Figure 2a–b*). Unlike Bdnf-e1 and -e2 -/- mice, Bdnf-e4 and -e6 -/- males do not show abnormal social behavior and Bdnf-e4 and -e6 -/- females breed successfully with no difficulties in pup survival.(*Maynard et al., 2016*) Quantitative PCR revealed that Bdnf-e1 and -e2 -/- females show a 50% reduction of *Oxt* transcripts at postnatal day 28 compared to WT females (*Figure 2a*; $F_{4,15}$= 3.249, p=0.0416). Reductions in *Oxt* transcripts were unique to Bdnf-e1 and -e2 -/- mice and were not observed in Bdnf-e4 and -e6 -/- females. Downregulation of *Oxt* transcripts in Bdnf-e1 -/- females was specific to development as adult mutants showed normal *Oxt* transcript levels compared to WT (t = 1.516, p>0.05). There were also no significant differences in hypothalamic *Oxtr* transcript levels between Bdnf mutants compared to WT (*Figure 2b*). To characterize the oxytocinergic system, we quantified the number of OXT-expressing cells in the PVN of WT and Bdnf-1 -/- postpartum mothers (*Figure 2c,f,i–k*). We found there were no significant differences in the size of the PVN (*Figure 2i*), total number of PVN cells (*Figure 2j*), or percentage of neurons expressing OXT (*Figure 2k*). Given that OXT neurons are engaged by pup exposure and retrieval (*Okabe et al., 2017*), we next determined whether Bdnf-e1 -/- females show changes in OXT neuron activation during displays of maternal behavior. We immunolabeled neurons for OXT and cFOS, a surrogate marker for activated neurons, in the PVN of WT and Bdnf-e1 -/- postpartum mothers two hours following pup retrieval testing (*Figure 2d,e,g,h*). We found no significant change in the percentage of OXT neurons labeled with cFOS (*Figure 2l*), suggesting that OXT neurons are equally recruited during pup retrieval in WT and Bdnf-e1 -/- postpartum mothers.

The receptor for BDNF, tropomyosin receptor kinase B (TrkB), is highly expressed in OXT neurons (*Figure 3a*), suggesting that BDNF may directly modulate OXT neuron function. To evaluate whether loss of BDNF signaling in OXT neurons contributes to impairments in maternal care, we ablated TrkB selectively in OXT-expressing neurons by crossing mice expressing Cre-recombinase under control of the endogenous *Oxt* promoter (Oxt^Cre) to mice expressing a floxed TrkB allele (TrkB^flox/flox; *Figure 3b*). To verify loss of TrkB selectively in OXT neurons, we performed immunohistochemistry for OXT and TrkB in Oxt^Cre, TrkB^flox/flox, and Oxt^Cre;TrkB^flox/flox mice (*Figure 3—figure supplement 1*). Oxt^Cre;TrkB^flox/flox mice (*Figure 3—figure supplement 1a*) showed reduced TrkB labeling in OXT neuron cell bodies compared to TrkB^flox/flox (*Figure 3—figure supplement 1b*) and Oxt^Cre (*Figure 3—figure supplement 1c*) mice. Although immunohistochemistry for TrkB had minimal background labeling (*Figure 3—figure supplement 1d*), high levels of synaptic staining in neighboring cells made it difficult to clearly visualize the absence of TrkB protein in OXT neurons. Therefore, to further validate successful recombination in OXT neurons, we performed single molecule fluorescent in situ hybridization using probes selective for *Oxt* as well as the floxed region of *Ntrk2* in TrkB^flox/flox mice, which includes the 5'UTR and exon S encoding the signal peptide (*Figure 3—figure supplement 2*, [*Baydyuk et al., 2011*]). As expected, Oxt^Cre;TrkB^flox/flox mice exhibited fewer nuclear messages for the floxed region of *Ntrk2* in OXT neurons (*Figure 3—figure supplement 2a*) compared to Oxt^Cre (*Figure 3—figure supplement 2b*) and TrkB^flox/flox mice (*Figure 3—figure supplement 2c*). This loss was selective to OXT neurons as *Ntrk2* expression was detected normally in surrounding cell types.

Unlike Bdnf-e1 -/- postpartum mothers, Oxt^Cre;TrkB^flox/flox postpartum mothers with selective deletion of TrkB in OXT neurons showed normal pup survival compared to control mice (*Figure 3c*).

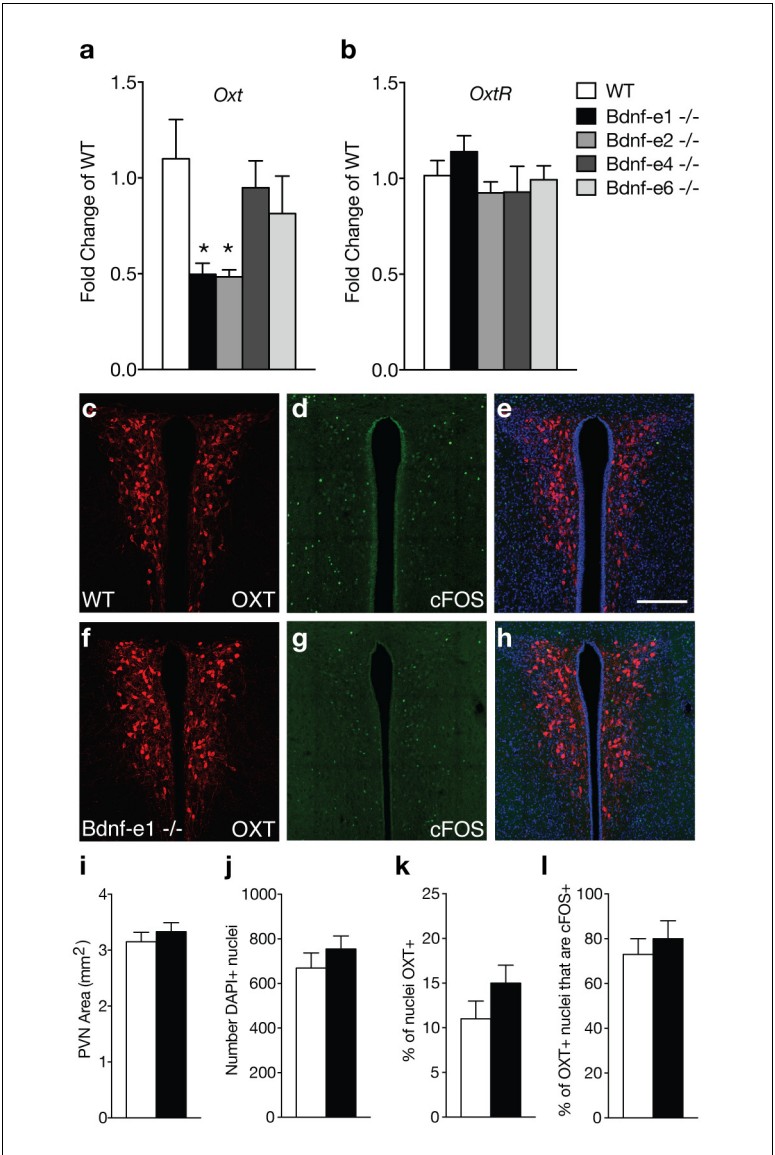

**Figure 2.** Oxytocin transcripts are reduced in Bdnf-e1 and -e2 -/- with impaired maternal care. (**a–b**) qPCR demonstrating relative expression levels of oxytocin (**a**) and oxytocin receptor (**b**) transcripts in the HYP of 4 week old Bdnf-e1, -e2, -e4, and -e6 female mice (n = 4–5 per genotype; 1-way ANOVAs with Bonferroni's multiple comparisons, p<0.05). *Oxt*, but not *Oxtr*, transcripts are reduced in Bdnf-e1 and -e2 -/- females. (**c–h**) Confocal *z*-projections of PVN from WT (**c–e**) and Bdnf-e1 -/- (**f–h**) postpartum mothers collected 2 hr following pup retrieval testing. Immunolabeling of OXT (**c, f**) cFOS (**d, g**), and merged images (**e, h**). (**i–l**) Quantification of PVN area (**i**), number of PVN cells (**j**), % of PVN cells expressing OXT (**k**), and percentage of OXT neurons expressing cFOS following pup retrieval (**l**) in WT and Bdnf-e1-/- postpartum mothers (n = 6 – 9 images per animal; n = 3 animals per genotype; student's *t*-test, Poisson regression, and binomial regression, respectively). There were no significant differences in PVN structure, OXT neuron number, or proportion of OXT neurons activated following pup retrieval. Data are means ± SEM; *p<0.05.
DOI: https://doi.org/10.7554/eLife.33676.005

However, similar to Bdnf-e1 -/- mothers, Oxt^Cre^; TrkB^flox/flox^ mothers showed a significant increase in latency to first pup retrieval (*Figure 3d*; t = 2.297, df = 25, p=0.0303) and a strong trend for increased incidences of moving away from pups without retrieving (*Figure 3e*). While analysis of global parenting style did not reveal significant reductions in parenting compared to control, a substantial proportion of Oxt^Cre^; TrkB^flox/flox^ mothers were 'non-parenters' and failed to engage with

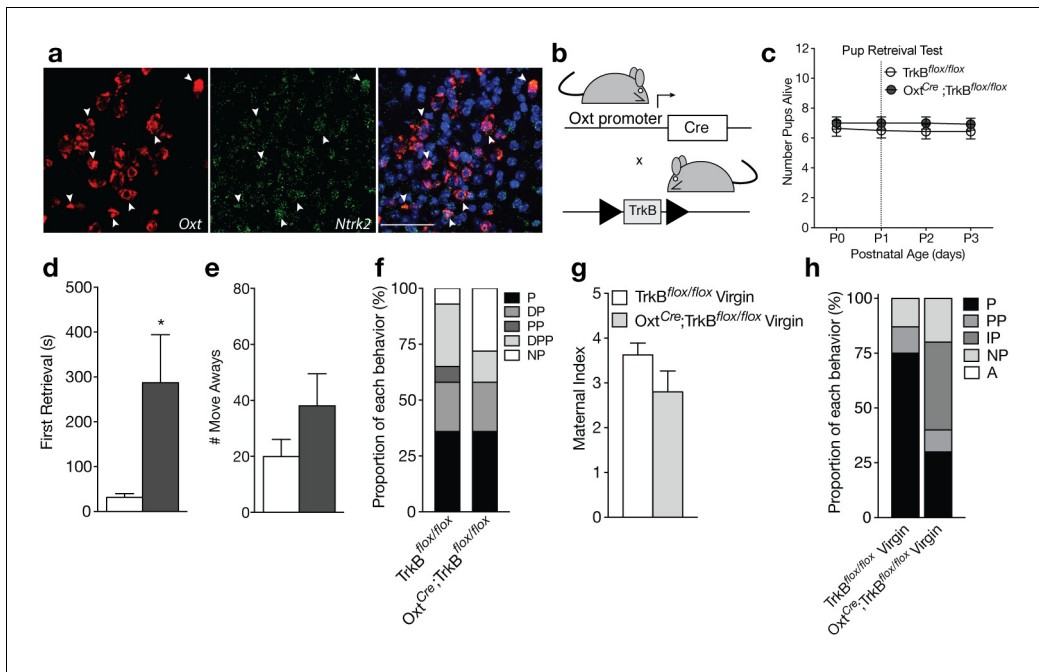

**Figure 3.** Loss of TrkB in OXT neurons leads to parenting deficits in postpartum mothers and virgin females. (**a**) Confocal *z*-projections of *Oxt* and *Ntrk2* transcripts in brain slices containing the PVN from an adult female visualized with RNAscope in situ hybridization. TrkB transcripts (green) are highly expressed in OXT neurons (red). (**b**) Breeding strategy used to obtain Oxt$^{Cre}$; TrkB$^{flox/flox}$ mice. (**c**) Average litter size over time for TrkB$^{flox/flox}$ and Oxt$^{Cre}$; TrkB$^{flox/flox}$ postpartum mothers exposed to the pup retrieval test one day post parturition. Mothers lacking TrkB in OXT neurons show normal pup survival compared to control (2-way ANOVA with mixed effect model; p>0.05). (**d**) Latency to first retrieval during pup retrieval test. Oxt$^{Cre}$; TrkB$^{flox/flox}$ postpartum mothers show increased latency to retrieve first pup (Student's *t*-test; p<0.05). (**e**) Number of times postpartum mothers move away from pups without retrieving. Oxt$^{Cre}$; TrkB$^{flox/flox}$ mothers show a strong trend for increased move-aways. (**f**) Proportion of Oxt$^{Cre}$; TrkB$^{flox/flox}$ and control mothers with different maternal types including parenting (P), disorganized parenting (DP), partial parenting (PP), disorganized partial parenting (DPP), and non-parenting (NP). There is no significant change in the proportion of Oxt$^{Cre}$; TrkB$^{flox/flox}$ mothers with parenting behavior compared to control (one-tailed Mann-Whitney-Wilcoxon rank sum test, p>0.05). However, a substantial proportion of Oxt$^{Cre}$; TrkB$^{flox/flox}$ mothers show non-parenting behaviors. (**g**) Maternal index for control and Oxt$^{Cre}$; TrkB$^{flox/flox}$ virgin females during foreign pup retrieval test. Oxt$^{Cre}$; TrkB$^{flox/flox}$ virgin females show a trend for reduced maternal index. (**h**) Proportion of Oxt$^{Cre}$; TrkB$^{flox/flox}$ and control mothers with different maternal types including parenting (P), partial parenting (PP), irregular parenting (IP), non-parenting (NP), and attack (A). There is a significant decrease in the proportion of Oxt$^{Cre}$; TrkB$^{flox/flox}$ virgins with parenting behavior compared to control (one-tailed Mann-Whitney-Wilcoxon rank sum test, p<0.05). Data are means ± SEM. (n = 14 TrkB$^{flox/flox}$ postpartum mothers; n = 14 Oxt$^{Cre}$; TrkB$^{flox/flox}$ postpartum mothers; n = 8 TrkB$^{flox/flox}$ virgins; n = 10 Oxt$^{Cre}$; TrkB$^{flox/flox}$ virgins; *p<0.05).

DOI: https://doi.org/10.7554/eLife.33676.006

The following figure supplements are available for figure 3:

**Figure supplement 1.** Ablation of TrkB protein in OXT neurons in Oxt$^{Cre}$; TrkB$^{flox/flox}$ mice.
DOI: https://doi.org/10.7554/eLife.33676.007
**Figure supplement 2.** Selective loss of TrkB mRNA in OXT neurons in Oxt$^{Cre}$; TrkB$^{flox/flox}$ mice.
DOI: https://doi.org/10.7554/eLife.33676.008
**Figure supplement 3.** Oxt$^{Cre}$ mice show normal maternal behavior.
DOI: https://doi.org/10.7554/eLife.33676.009

relocated pups (*Figure 3f*). While Oxt$^{Cre}$; TrkB$^{flox/flox}$ virgin females did not attack foreign pups as observed with Bdnf-e1 -/- virgins, a significant percentage of Oxt$^{Cre}$; TrkB$^{flox/flox}$ virgins failed to parent (*Figure 3h*; Mann-Whitney-Wilcoxon rank sum test, W = 21.5, corrected p=0.04236) and showed trends for reduced nurturing behavior (*Figure 3g*). Interestingly, Oxt$^{Cre}$; TrkB$^{flox/flox}$ females showed no evidence of mating abnormalities and male breeders paired with these females did not sustain

genitalia injuries. To rule out the possibility that maternal care deficits in $Oxt^{Cre}$; TrkB$^{flox/flox}$ might be driven by presence of the $Oxt^{Cre}$ allele, we also tested $Oxt^{Cre}$ virgin females on the pup retrieval test (*Figure 3—figure supplement 3*). Compared to WT virgin females, $Oxt^{Cre}$ virgin females showed normal maternal behaviors including pup nurturing (*Figure 3—figure supplement 3a*) and retrieval (*Figure 3—figure supplement 3c*). $Oxt^{Cre}$ females also exhibited minimal abnormal parenting behaviors (*Figure 3—figure supplement 3b*) and the majority of $Oxt^{Cre}$ mice engaged in robust parenting displays (*Figure 3—figure supplement 3d*). These results suggest that direct loss of BDNF-TrkB signaling in OXT neurons leads to altered maternal behavior.

## Translatome profiling of OXT neurons in sexually mature females

Magnocellular and parvocellular neurons that release OXT are highly plastic in response to prolonged activation (*Theodosis, 2002*). However, the mechanisms underlying OXT neuron plasticity are not fully understood due to limited knowledge regarding the molecular identity of OXT neurons, which has been difficult to characterize due to the intermingled distribution of this population within the hypothalamus, particularly the paraventricular and supraoptic nuclei. To address this challenge and gain mechanistic detail into how BDNF-TrkB signaling might be impacting OXT neuron function and plasticity to modulate female-typical social behaviors, we used translating ribosome affinity purification (TRAP) followed by RNA sequencing (RNA-seq) to characterize actively translating RNAs in hypothalamic neurons expressing OXT. We began by crossing $Oxt^{Cre}$ mice to a Cre-dependent ribosome-tagged mouse (RiboTag mouse, referred to as Rpl22$^{HA}$) to allow for HA-tagging of ribosomes under control of the endogenous *Oxt* promoter (*Figure 4a*). Ribosomes tagged selectively in OXT neurons were immunoprecipitated (IP) from total hypothalamic homogenate (Input) using an anti-HA antibody. Actively translating mRNAs were isolated from IP fractions and total mRNA was isolated from Input fractions.

Cell type-specific expression of the RiboTag allele in OXT neurons was confirmed by qPCR analysis showing significant enrichment (>100 fold) of *Oxt* expression in IP compared to Input (*Figure 4b*). Following RNA amplification and library construction, we performed high-throughput RNA-seq on Input and IP RNA fractions to generate a comprehensive molecular profile of genes enriched and depleted in OXT neurons. Among the 22,472 expressed genes (at RPKM > 0.1, *Supplementary file 1*—Table S1), we identified 1670 differentially expressed between Input and IP RNA fractions at FDR < 1%, including 1129 genes with fold changes greater than 2 (*Supplementary file 1*—Table S2, *Figure 4c*). Differential analysis confirmed significant enrichment (IP/Input) of *Oxt* transcripts (59-fold increase, t = 16.6, p=2.94 $\times$ 10$^{-11}$) and significant depletion of transcripts for *Agrp*, *Cartpt*, and *Pmch* (3.8, 3.6, and 4.2 fold decreases, p$_{adj}$ = 0.015, 8.88 $\times$ 10$^{-5}$, and 2.31 $\times$ 10$^{-5}$ respectively), genes enriched in other hypothalamic nuclei, including the arcuate and lateral hypothalamus, respectively. *Supplementary file 1*—Table S1 includes *Bdnf* and *Ntrk2*, the gene encoding TrkB, which is highly expressed throughout the hypothalamus, including in OXT neurons. To independently validate our RNA-seq results, we confirmed differential expression of a subset of enriched and de-enriched genes in OXT neurons (*Figure 4—figure supplement 1*). Using qPCR, we verified significant enrichment of *Myo5a*, *Ank2*, *Peg3*, and *Kmt2a* and de-enrichment of *Agrp* and *Cartpt* in OXT IP compared to Input samples (*Figure 4—figure supplement 1a*). Using single molecule fluorescent in situ hybridization in adult WT virgin females, we further demonstrated enrichment of *Kmt2a* (*Figure 4—figure supplement 1b*), *Peg3* (*Figure 4—figure supplement 1c*), and *Ank2* (*Figure 4—figure supplement 1d*) transcripts in OXT neurons.

Highly enriched translating mRNA species in OXT neurons presumably represent the most functionally relevant genes for this cell type in the hypothalamus. To discover which classes of mRNAs are preferentially expressed in female OXT hypothalamic neurons, we performed GO enrichment analysis on the subset of the 756 genes (with Entrez IDs) more highly expressed in Oxt neurons compared to total hypothalamic homogenate and also on the 813 Entrez genes more highly expressed in homogenate tissue (*Figure 4d*; *Supplementary file 1*—Table S3). Analysis with the cellular component category showed that OXT-enriched mRNAs encode proteins critical for structural plasticity, including cytoskeletal and postsynaptic organization. Analysis with the molecular function category revealed genes that encode signaling molecules critical for binding to actin, microtubules, GTPases, calmodulin, and syntaxin-1, suggesting robust expression of proteins that regulate synaptic structure and exocytosis in OXT neurons. Reassuringly, pathways relevant for gliogenesis, myelin, and

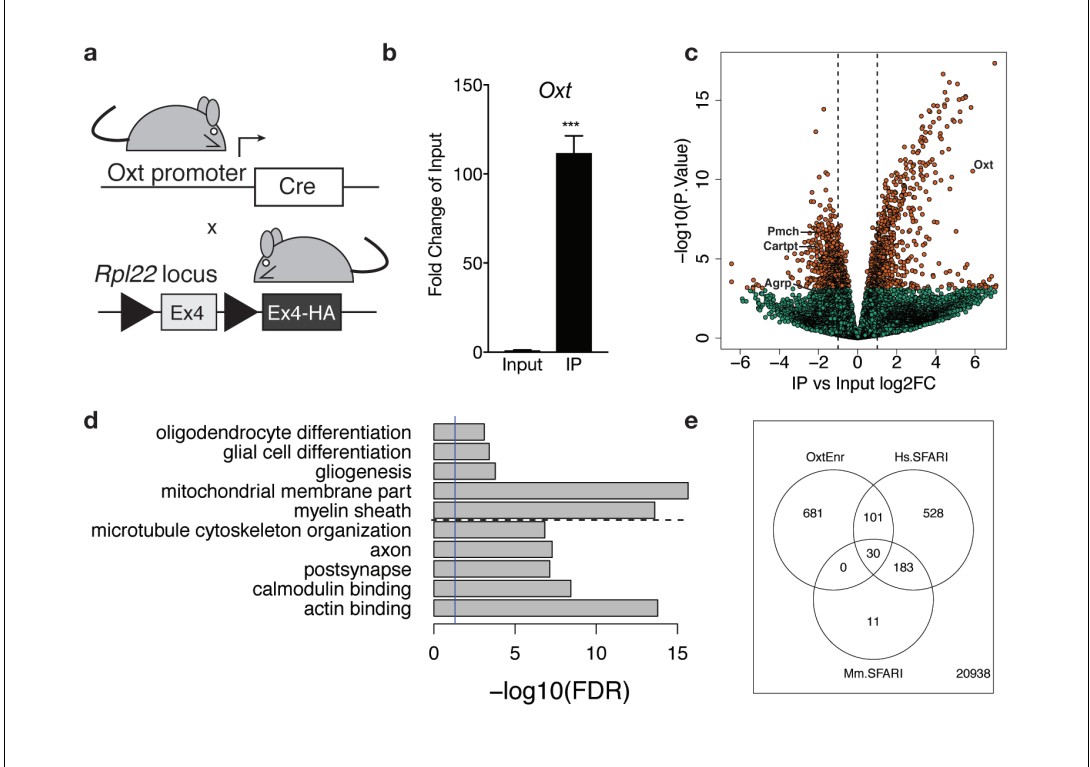

**Figure 4.** Identification of a unique molecular profile for OXT neurons using ribosomal tagging. (**a**) Locus of the ribosomal protein *Rpl22* in the Ribotag mouse and breeding strategy used to obtain Oxt$^{Cre}$; Rpl22$^{HA}$ mice. (**b**) qPCR analysis validating significant enrichment of *Oxt* transcripts in IP compared to Input fractions. (**c**) Volcano plot of RNA-seq results with IP versus Input log$_2$ fold change against –log$_{10}$ p-value. Outer lines are 2-fold enriched/depleted genes. Red dots represent genes that are significantly different (<0.05) in IP versus Input. Green dots represent non-significant genes. A subset of differentially enriched marker genes is highlighted (*Oxt, Agrp, Cartpt, Pmch*). (**d**) Representative gene ontology (GO) terms in the cellular component, molecular function and biological processes categories for genes enriched and de-enriched in OXT neurons. Solid vertical line indicates p=0.05 and dotted horizontal line separates downregulated (top) and upregulated (bottom) pathways. (**e**) Venn diagram showing overlap of differentially expressed genes (DEGs) in OXT neurons with mouse (Mm) and human (Hs) genes implicated in autism spectrum disorder by the Simons Foundation Autism Research Initiative (SFARI). Data are means ± SEM. (n = 3 Input, n = 3 IP; 5 Oxt$^{Cre}$; Rpl22$^{HA}$ hypothalami pooled per sample).

DOI: https://doi.org/10.7554/eLife.33676.010

The following figure supplements are available for figure 4:

**Figure supplement 1.** Validation of select genes enriched or de-enriched in OXT neurons.

DOI: https://doi.org/10.7554/eLife.33676.011

**Figure supplement 2.** Validation of Ovation RNA-seq V2 system with SoLo RNA-seq system.

DOI: https://doi.org/10.7554/eLife.33676.012

oligodendrocyte differentiation contained genes more highly expressed in homogenate tissue compared to OXT neurons.

To confirm that RNA amplification prior to library construction did not bias sequencing data, we also generated low input RNA-seq libraries directly from Input and IP RNA fractions using the Nugen Ovation SoLo RNA-seq System, which produces strand-specific, rRNA depleted libraries from 10 pg-10ng of RNA. Replication of differentially expressed genes (DEGs) using the SoLo kit was very high. Approximately 99% of FDR-significant genes from RNA amplified samples were in the same direction as those from SoLo libraries, of which 92% were marginally significant (p<0.05) (***Figure 4—figure supplement 2***; ***Supplementary file 1***—Table S2). To identify a 'neuronal translation' signature of genes that are enriched on translating ribosomes in neurons independent of cell-type, we compared our Oxt Input and IP fractions to publicly available TRAP data from Ntsr1+ cortical layer VI neurons (***Nectow et al., 2017***) and Input and IP fractions obtained from cortistatin (*Cort*) inhibitory interneurons (***Supplementary file 1***—Table S2). We identified 14 genes (FDR < 0.01 in Oxt and p<0.01 in the other two cell types) enriched on actively translating ribosomes in OXT neurons as well as in excitatory and inhibitory neurons (*Etl4, Kalrn, Actn4, Copa, Myo5b, Cdc42bpb, Phrf1, Gtf3c1,*

*Srrm1, Nefm, Srcin1, Mapk8ip3, Fasn, Arhgef11*). All three cell types were de-enriched for a set of 50 genes, including *Apoe* and *Mog*, genes enriched in astrocytes and oligodendrocytes, respectively.

We also compared our Oxt IP fraction to combined Ntsr1+ and Cort IP fractions to identify genes differentially expressed between hypothalamic OXT neurons and classical excitatory and inhibitory cell populations in the cortex (*Supplementary file 1*—Table S2, Table S4). An unfiltered list of genes highly specific for OXT neurons compared to Ntsr1+ and Cort neurons is found in *Supplementary file 1*—Table S4. The top 5 differentially expressed genes between these neuronal cell types were *Irs4*, *Hcrt*, *Oxt*, *Peg10*, and *Mef2c*. Gene ontology (GO) enrichment analysis on the FDR < 1% Oxt IP versus Ntsr1+ and Cort IP genes showed that OXT neurons express significantly different classes of mRNAs compared to other cell types (*Supplementary file 1*—Table S5). For example, analysis with the molecular function category revealed that OXT neurons are enriched for genes encoding G-protein coupled receptor activity and peptide receptor activity compared to Ntsr1+ and Cort neurons. Conversely, OXT neurons are de-enriched for genes encoding voltage gated ion channel activity compared to Ntsr1+ and Cort cell populations.

Given the central role of oxytocin in regulating social behavior, we examined whether genes enriched in female OXT neurons (e.g. more highly expressed than homogenate tissue and FDR < 1%) overlap with genes implicated in autism spectrum disorder (ASD) by the Simons Foundation Autism Research Initiative (SFARI). We first found strong enrichment among the 224 genes with mouse Genetic Animal Models of ASD - 30 of these genes (13%, OR = 4.24, p=6.62e-10) were significantly more highly expressed in OXT neurons that homogenate tissue (*Supplementary file 1*—Table S6). We further found strong enrichment of genes in the SFARI Human Gene Module using the subset of mouse-expressed genes with human homologs (N = 14,769) – 131 of the 842 expressed Human SFARI genes were enriched in OXT neurons (15.5%, OR = 4.0, p<2.2e-16). (*Figure 4e*; *Supplementary file 1*—Table S7). Many of these genes encode proteins that modulate synaptic function and plasticity, suggesting a link between OXT neuron adaptability and social behavior.

## Disruption of BDNF alters gene expression in OXT neurons

To delineate how perturbations in BDNF signaling impact OXT-enriched genes to modulate female-typical social behavior, we next compared the translatome of OXT neurons in control versus Bdnf-e1 mutant mice. Due to difficulties in obtaining Bdnf-e1; Oxt$^{Cre}$; Rpl22$^{HA}$ triple crossed mice (*Bdnf* and *Oxt* are both located on chromosome two resulting in infrequent recombination of the mutant allele with the Cre transgene and RiboTag reporter), we switched to a viral strategy and sought to optimize expression profiling in OXT neurons from individual, instead of pooled, samples. We generated Bdnf-e1; Oxt$^{Cre}$; tdTom mice and utilized an adeno-associated virus (AAV) that initiates Cre-inducible RiboTag expression using a double floxed inverted open reading frame (DIO) approach (*Sanz et al., 2015*) (*Figure 5a*). Viral delivery into the PVN (*Figure 5b*) had two additional advantages: (1) it ensured profiling of neurons that were actively utilizing the *Oxt* promoter in adulthood and (2) it allowed for targeting of OXT populations selectively in the PVN as opposed to the supraoptic nucleus (SON). We isolated IP RNA from control (Bdnf-e1 +/+; Oxt$^{Cre}$; tdTom) and mutant (Bdnf-e1 -/-; Oxt$^{Cre}$; tdTom) virgin female mice infused with AAV-DIO-Ribotag and directly generated RNA-seq libraries from each individual mouse using the Nugen Ovation SoLo RNA-seq System.

Differential expression analysis accounting for sample variation and differences in transfection or enrichment efficiency (via exonic mapping rate and *Oxt* expression itself) revealed 100 differentially expressed genes between control IP vs. Bdnf-e1 -/- IP samples at FDR < 10% which was controlled by p<0.00038. (*Figure 5c*; *Supplementary file 1*—Table S8). Importantly, DEGs between Oxt IP vs. Oxt Input in WT females did not overlap with DEGs between control IP and Bdnf-e1 -/- IP samples, suggesting efficient immunoprecipitation of ribosomes in OXT neurons. (*Figure 5—figure supplement 1*). Of particular interest, we identified significant elevation in *Gabra2*, a subunit for the GABA$_A$ receptor, which is known to mediate inhibition onto OXT neurons and shows expression modulation through BDNF signaling (*Choe et al., 2015*; *Hewitt and Bains, 2006*). Furthermore, Bdnf-e1 -/- OXT neurons exhibited downregulation of *Cckar*, a gene encoding the cholecystokinin A receptor, which is critical for female mating behavior (*Xu et al., 2012*). Changes in the levels of *Gabra2* and *Cckar* between control and Bdnf-e1 -/- OXT neurons were validated by fluorescent in situ hybridization using RNAscope (*Figure 5e,f*). Unexpectedly, a number of the differentially expressed genes in OXT

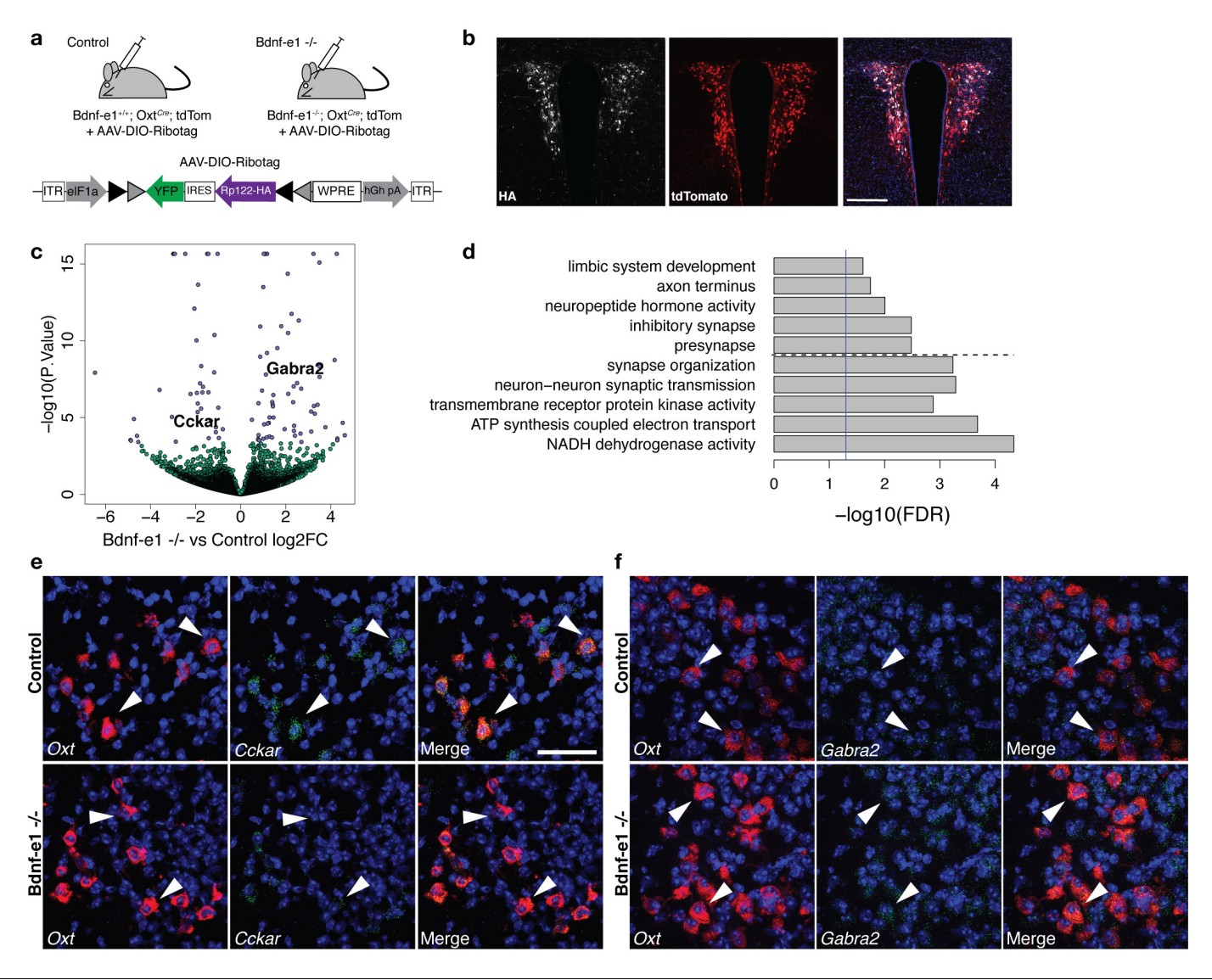

**Figure 5.** Perturbations in BDNF signaling impact gene pathways critical for plasticity in OXT neurons. (a) Strategy for TRAP in OXT neurons from control and Bdnf-e1 -/- mice (n = 3 per genotype) using AAV-DIO-RiboTag and RNA-seq. (b) Confocal image showing HA (white) and tdTom (red) expression in hypothalamic sections containing the PVN of Bdnf-e1; Oxt^Cre; tdTom mice injected bilaterally with AAV-DIO-Ribotag. Scale bar is 200 μm. (c) Volcano plot of RNA-seq results with Bdnf-e1 -/- IP vs. control IP log₂ fold change against -log₁₀ p-value. Blue dots represent genes that are significantly different in Bdnf-e1 -/- IP vs. control IP, including *Cckar* and *Gabra2*. Green dots represent non-significant genes. (d) Representative gene ontology (GO) terms in the molecular function, biological processes, and cellular component categories for genes enriched and de-enriched in OXT neurons following disruption of BDNF signaling. Solid vertical line indicates p=0.05 and dotted horizontal line separates downregulated (top) and upregulated (bottom) pathways. (e) Confocal *z*-projections of *Oxt* and *Cckar* transcripts in brain slices containing the PVN from adult control and Bdnf-e1 -/- females visualized with RNAscope in situ hybridization. *Cckar* transcripts (green) are enriched in *Oxt*-expressing neurons (red) in control, but not Bdnf-e1 -/- females. (f) Confocal *z*-projections of *Oxt* and *Gabra2* transcripts in adult PVN of control and Bdnf-e1 -/- females visualized with RNAscope in situ hybridization. *Gabra2* transcripts (green) co-localize with *Oxt* transcripts (red) in control females and appear elevated in Bdnf-e1 -/- females. Scale bar is 50 μm.

DOI: https://doi.org/10.7554/eLife.33676.013

The following figure supplement is available for figure 5:

**Figure supplement 1.** DEGs in Oxt Input vs. Oxt IP do not overlap with DEGs in control IP vs. Bdnf-e1 -/-IP.
DOI: https://doi.org/10.7554/eLife.33676.014

neurons between control and Bdnf-e1 -/- were mitochondrial genes, suggesting that BDNF disruption leads to dysregulation in oxidative phosphorylation systems that mediate cellular energy and metabolism. This is consistent with previous literature demonstrating a neuroprotective role for BDNF in mitigating metabolic defects and promoting cellular stress resistance during plasticity (*Raefsky and Mattson, 2017*; *Xu et al., 2018*).

GO analysis with the biological process, molecular function, and cellular component categories identified gene sets significantly different between control and Bdnf-e1 -/- OXT neurons as encoding proteins that are known to function at synapses, axons, and the respiratory chain in mitochondria (*Figure 5d*; *Supplementary file 1*—Table S9). For example, GO terms significantly different in the molecular function category included NADH (nicotinamide adenine dinucleotide) dehydrogenase activity, transmembrane receptor protein tyrosine kinase activity, and neuropeptide hormone activity (*Figure 5d*). Differentially expressed gene sets between control and Bdnf-e1 -/- OXT neurons were also enriched in development-related GO terms for biological processes such as axon development, neuronal differentiation and migration, and synaptic assembly and transmission (*Supplementary file 1*—Table S9). We also examined whether DEGs between control and Bdnf-e1 -/- OXT neurons overlap with SFARI genes implicated in ASD. Here we found significant enrichment with the Genetic Animal Model ASD genes, specifically *Reln*, *Nrp2*, *Erbb4*, *Gad1*, and *Pou3f2*, with those genes dysregulated in OXT neurons following disruption of BDNF signaling (p=0.0036, OR = 5.21, *Supplementary file 1*—Table S10) with only suggestive enrichment of the more general Human SFARI ASD genes (p=0.067, OR = 2.0) which contains the above five mouse model genes plus *Avp* and *Chrnb3*. Taken together, these results suggest that perturbations in hypothalamic BDNF-TrkB signaling lead to dysregulation of plasticity mechanisms in OXT neurons that may impact female social behavior.

## Discussion

While a casual role for BDNF in regulating male-typical social behaviors has been well-established, the contribution of BDNF signaling to female-typical social behaviors has remained unexplored. Here we demonstrate a novel role for BDNF in modulating maternal care and sexual receptivity in female mice. Furthermore, we provide evidence that BDNF-TrkB signaling in hypothalamic OXT neurons contributes to maternal behavior. We define a molecular profile for OXT neurons and demonstrate that perturbations in BDNF-TrkB signaling impact gene expression in OXT neurons, potentially influencing female-typical social behaviors. Our studies identify hypothalamic BDNF as a broad modulator of sex-specific social behaviors and elucidate new activity-dependent synaptic plasticity pathways critical for OXT neuron function.

### BDNF impacts female-typical social behaviors

The majority of literature surrounding BDNF in the context of maternal care has focused on offspring. A large body of work has demonstrated that increased prenatal stress or reduced maternal care leads to downregulation of BDNF in developing offspring and subsequent deficits in adult behavior (*Branchi et al., 2013*; *Cirulli et al., 2009*; *Hill et al., 2014*). However, no studies have investigated the reverse relationship to ask whether BDNF levels in mothers play a causal role in regulating displays of maternal behavior. This is especially surprising given that BDNF is well-established as a robust modulator of male-typical social behavior, particularly aggression. Heterozygous mice with a 50% reduction of global BDNF show enhanced aggression and alterations in serotonin signaling (*Lyons et al., 1999*). Furthermore, we have previously shown that selective disruption of BDNF from promoters I or II, but not IV or VI, leads to significant loss of BDNF in the hypothalamus and elevated aggression and mounting behavior during fighting (*Maynard et al., 2016*). Gaps in understanding how BDNF controls female social behavior are also surprising given that the BDNF gene contains an estrogen response element (*Sohrabji et al., 1995*) and estrogen robustly increases BDNF expression and impacts dendritic spine plasticity to modulate behaviors such as cognition (*Luine and Frankfurt, 2013*). These studies suggest important sexually dimorphic effects of BDNF on neural circuitry and behavior that have been relatively unexplored.

Akin to social deficits in Bdnf-e1 and -e2 -/- males, we find that selective disruption of promoter I and II-derived BDNF in females leads to significant impairments in sex-typical social behaviors, including pup retrieval and mating behavior. While WT postpartum females exhibit stereotyped

patterns of maternal care in which they systematically retrieve and nurture pups (**Lonstein and De Vries, 2000**), Bdnf-e1 and -e2 -/- dams show disorganized parenting behavior and decreased pup survival. Furthermore, unlike WT virgins that show affiliative behaviors towards foreign pups (**Svare and Mann, 1981**), Bdnf-e1 -/- virgins find foreign pups aversive and ignore or attack them. Bdnf-e1 -/- females also display reduced sexual receptivity towards WT males resulting in male genitalia injuries, disrupted mating patterns, and decreased probability of conception. Similar to Bdnf-e1 and Bdnf-e2 -/- males, Bdnf-e1 and -e2 -/- females show differences in phenotype severity with Bdnf-e1 -/- females having stronger phenotypes. For example, Bdnf-e1 -/- postpartum females exhibit greater pup loss compared to Bdnf-e2 -/- postpartum females. We speculate that decreased pup survival for Bdnf-e1 -/- dams may result from a combination of parental neglect, infanticide, and inconsistencies in nursing behavior. Future experiments should distinguish between these possibilities and explore the role of BDNF-TrkB signaling in regulation of lactation, which is modulated by neuropeptides such as oxytocin (**Nishimori et al., 1996**). Given that promoter I is more highly sensitive to neural activity than promoter II (**Timmusk et al., 1993**), it is possible that phenotype severity in Bdnf-e1 -/- females may be accounted for by loss of activity-induced BDNF that is critical for synaptic plasticity. Furthermore, Bdnf-e2 -/- mice show compensatory downregulation of *Bdnf* exon I transcripts (**Maynard et al., 2016**), which may drive their exhibited deficits in maternal care.

## Relationship between BDNF signaling, oxytocin, and maternal behavior

Bdnf-e1 and -e2 mutants show a 50% reduction of BDNF protein in the hypothalamus, suggesting that promoters I and II are heavily used in this brain region (**Maynard et al., 2016**). BDNF and its receptor TrkB are highly expressed in several hypothalamic nuclei, including the paraventricular nucleus, a primary site of magnocellular and parvocellular neurons that secrete oxytocin (**Dölen, 2015**; **Smith et al., 1995**). Oxytocin is a neuropeptide with well-established roles in modulating sex-typical social behaviors, including maternal behaviors such as pup retrieval (**Marlin et al., 2015**; **Pedersen et al., 1982**). We find that disruption of BDNF production from promoters I and II leads to transient decreases in *Oxt* gene expression in Bdnf-e1 and -e2 -/- females before sexual maturity. Early dysregulations in *Oxt* expression may be indicative of developmental impairments in the oxytocinergic system that contribute to mating and parenting deficits in adult females. Indeed, manipulating levels of OXT during early postnatal development has long-lasting effects on social behaviors into adulthood (**Bales and Carter, 2003**; **Peñagarikano et al., 2015**; **Yamamoto et al., 2004**). Interestingly, sexually mature Bdnf-e1 -/- females show normal *Oxt* expression compared to WT and do not exhibit deficits in the number or activation of OXT neurons during pup retrieval. These findings suggest that BDNF may not regulate OXT neuron survival or recruitment, but may instead have more subtle effects on synaptic plasticity or gene regulation in OXT neurons. Our molecular profiling data support this hypothesis and provide evidence that BDNF may modulate the morphology and activity of OXT neurons to influence the location and timing of OXT release.

Importantly, loss of TrkB in OXT neurons partially recapitulates maternal care deficits in Bdnf-e1 -/- females, suggesting that BDNF-TrkB signaling in OXT neurons is sufficient to modulate female-typical social behavior. However, as phenotypes in Oxt$^{Cre}$; TrkB$^{flox/flox}$ females were milder than those observed in Bdnf-e1 -/- females, it is likely that BDNF-TrkB signaling in other hypothalamic populations associated with parental behavior, such as the medial preoptic area (MPO) and antero-ventral periventricular nucleus (AVPV) (**Scott et al., 2015**; **Wu et al., 2014**), contribute to the regulation of female-typical social behaviors. Indeed, a monosynaptic circuit linking dopamine neurons in the periventricular hypothalamus to OXT neurons in the PVN was recently identified and shown to control OXT neuron secretion and maternal care (**Scott et al., 2015**). As BDNF plays an important role in the survival and function of dopamine neurons (**Altar et al., 1992**; **Hyman et al., 1991**; **Lobo et al., 2010**; **Loudes et al., 1999**), future studies should investigate whether BDNF-TrkB signaling also acts in this circuit to impact female-typical social behaviors. The milder phenotypes in Oxt$^{Cre}$; TrkB$^{flox/flox}$ mice compared to Bdnf-e1 -/- mice may also be due to differences in penetrance as Oxt$^{Cre}$; TrkB$^{flox/flox}$ and Bdnf-e1 -/- mice were on different genetic backgrounds (mixed C57B/6J and 129 versus C57B/6J, respectively). Indeed, genetic background can significantly influence performance on the pup retrieval test (**Carlier et al., 1982**), which is supported by our data indicating that baseline maternal behavior may differ between virgin female controls in *Figure 1p* and *Figure 3h*. Finally, it is possible that phenotypes were milder in Oxt$^{Cre}$; TrkB$^{flox/flox}$ mice compared to Bdnf-e1 -/- mice as a small number of OXT neurons may retain some TrkB. This could be due to in

complete recombination of the TrkB$^{flox/flox}$ allele or absence of Cre recombinase. As Cre activity was previously observed in 92% of oxytocin-expressing neurons in the PVN (*Wu et al., 2012*), we expect less than 10% of OXT cells to express TrkB.

## Molecular profile for OXT neurons in sexually mature females

Establishing a molecular profile for OXT neurons has been hindered by the intermingled distribution of this population within the hypothalamus, which poses a significant challenge for achieving cell-type specific gene expression. Over a decade ago, single-cell RT-PCR was used to identify cell-specific mRNAs expressed in magnocellular neurons selectively expressing oxytocin or vasopressin (*Yamashita et al., 2002*). While the authors identified 48 previously unknown genes to be expressed in magnocellular neurons in the lactating rat, the microarray approach did not allow for characterization of global gene expression. Using a translating ribosome affinity purification approach combined with RNA-sequencing, we identified ~1700 actively translated genes differentially expressed in OXT neurons compared to total hypothalamic homogenate, including ~1000 genes with a greater than 2-fold change in either direction. Gene ontology analysis revealed that many OXT-enriched genes encode proteins critical for structural and functional plasticity that act in pathways modulating synaptic and cytoskeletal organization as well as calmodulin and actin binding. As OXT neurons show remarkable morphological, electrophysiological, and synaptic remodeling in response to environmental events such as parturition and lactation (*Theodosis, 2002*), our results suggest a dynamic gene transcription program that supports rapid activity-induced, experience-dependent plasticity. For example, OXT neurons are enriched in *Ank2*, a gene encoding ankyrin 2, which plays a key role in synaptic organization, particularly the targeting and regulation of ion channels (*Kline et al., 2014*; *Koch et al., 2008*). OXT neurons are also enriched in *Kmt2a*, a gene encoding histone-lysine N-methyltransferase 2A, which is a global regulator of gene transcription and an epigenetic regulator of complex behaviors (*Jakovcevski et al., 2015*). Interestingly, we also saw enrichment of paternally-expressed gene 3, or *Peg3*, an imprinted gene that is expressed exclusively from the paternal allele and regulates maternal behavior (*Li et al., 1999*). Future studies should identify the molecular profile of OXT neurons in males to determine sexually dimorphic gene expression that may underlie male and female-typical social behaviors.

Identification of a molecular profile for OXT neurons may provide new insights into how the oxytocin system can be modulated to improve social behavior, especially in the context of autism spectrum disorder (ASD), which is characterized by deficits in social communication and interaction. Indeed, we find that differentially expressed genes in OXT neurons overlap with both mouse and human genes that have been implicated in ASD by the Simons Foundation Autism Research Initiative. Many of these genes encode proteins that are critical for structural and functional plasticity and could represent novel targets for enhancing OXT release. Given that OXT is of great interest for therapeutic intervention but does not easily cross the blood brain barrier (*Landgraf et al., 1979*; *McEwen, 2004*), the molecular profile of OXT neurons presented here provides a valuable resource for probing gene pathways critical for OXT expression and regulation. One limitation of our study is that we do not distinguish between magnocellular and parvocellular neurons, which have different neuronal projections and functions (*Dölen, 2015*). While it is likely that OXT neurons can be classified into at least four different subtypes (*Althammer and Grinevich, 2017*; *Romanov et al., 2017*), future experiments using retrograde expression of the RiboTag allele to label specific OXT projections may begin to parse out molecular profiles for functionally distinct classes of OXT neurons.

## BDNF modulates plasticity genes in OXT neurons

BDNF is a robust modulator of activity-dependent gene expression and synaptic plasticity (*Berton et al., 2006*; *Lu, 2003*). To determine how perturbations in BDNF signaling impact OXT-enriched genes to potentially influence female-typical social behaviors, we compared the translatomes of OXT neurons from control and Bdnf-e1 -/- mice. We found ~100 genes differentially expressed in OXT neurons following disruption of promoter I-derived BDNF. Differentially expressed genes in Bdnf-e1 -/- OXT neurons were enriched in pathways critical for morphological and synaptic plasticity. Gene ontology analysis revealed that many of these genes encode proteins that function in mitochondrial pathways important for cellular energy and adaptive responses to neural activity. This is consistent with previous reports showing that BDNF stimulates mitochondrial biogenesis in

hippocampal neurons (*Cheng et al., 2012*) and can enhance synaptic transmission by arresting mito-chondria at active synapses to fuel dendritic spine dynamics and synaptic plasticity (*Li et al., 2004*; *Raefsky and Mattson, 2017*; *Su et al., 2014*).

We also found significant elevation of *Gabra2*, a gene encoding a subunit for the GABA$_A$ receptor. Plasticity at inhibitory synapses is strongly modulated by expression, localization, and function of GABA$_A$ receptors (*Mele et al., 2016*), which mediate inhibition onto magnocellular and parvocellular neurons and play a key role in their adaptive responses following stimulation (*Bali and Kovács, 2003*; *Lee et al., 2015*). Elevation of *Gabra2* following loss of BDNF signaling is consistent with previous reports demonstrating that BDNF is a robust modulator of synaptic inhibition and GABA$_A$ receptors (*Brünig et al., 2001*; *Jovanovic et al., 2004*; *Tanaka et al., 1997*) and can reduce inhibitory synaptic drive on neuroendocrine cells by decreasing GABA$_A$ surface expression (*Hewitt and Bains, 2006*). BDNF has also been shown to decrease inhibition of vasopressin neurons through downregulation of GABA$_A$ receptor signaling (*Choe et al., 2015*). Taken together, these findings suggest that disruption of hypothalamic BDNF may lead to increased inhibition onto OXT neurons that could alter the timing or levels of OXT release to impact social behaviors. Interestingly, we also saw substantial downregulation of *Cckar*, a gene encoding the cholecystokinin A receptor. While *Cckar* is associated with feeding and metabolism (*Baile et al., 1983*), it is also essential for sexual receptivity in females (*Xu et al., 2012*). Similar to Bdnf-e1 -/- females, *Cckar* null females show reduced sexual receptivity, suggesting that BDNF and cholecystokinin signaling may interact in OXT neurons to control female sexual behavior. As estrogen induces both *Bdnf* and *Cckar* expression (*Luine and Frankfurt, 2013*; *Xu et al., 2012*), this may point to a sexually-dimorphic pathway in OXT neurons that influences female-specific social behaviors. An important future direction will be to complete parallel molecular profiling experiments in Oxt$^{Cre}$; TrkB$^{flox/flox}$ females to better understand the direct effects of BDNF-TrkB signaling on OXT neuron function.

In conclusion, we demonstrate that BDNF-TrkB signaling in oxytocin neurons, and likely additional hypothalamic populations, plays a causal role in modulating female-typical social behavior. We identify OXT neurons as enriched in genes important for structural and functional plasticity, and show that perturbations in BDNF signaling lead to disruption of OXT neuron gene expression that may impact maternal behavior in females.

# Materials and methods

**Key resources table**

| Reagent type (species) or resource | Designation | Source or reference | Identifiers | Additional Information |
|---|---|---|---|---|
| Genetic reagent (Mus musculus) | Bdnf-e1; Bdnf-e2 mice | PMID: 26585288 | | |
| Genetic reagent (Mus musculus) | *Oxt*-Cre mice | Jackson Labs | RRID:IMSR_JAX:024234 | |
| Genetic reagent (Mus musculus) | tdTomato mice | Jackson Labs | RRID:IMSR_JAX:007914 | |
| Genetic reagent (Mus musculus) | *Ntrk2* flox/flox mice | PMID: 18511296 | | |
| Genetic reagent (Mus musculus) | *Cort*-Cre mice | Jackson Labs | RRID:IMSR_JAX:010910 | |
| Genetic reagent (Mus musculus) | RPL22-HA mice | Jackson Labs | RRID:IMSR_JAX:011029 | |
| Genetic reagent (Adeno-assoicated virus) | AAV1-DIO-RPL22$^{HA}$-GFP | PMID: 25855171 | | |
| Antibody | anti-mouse HA | Covance | RRID:AB_291262 | |
| Antibody | goat anti-mouse Alexa 647 | Thermo-Fisher Scientific | RRID:AB_141693 | |
| Antibody | anti-mouse OXT | PMID: 3880813 | | |
| Antibody | anti-rabbit cFOS | Millipore | RRID:AB_2631318 | |

*Continued on next page*

*Continued*

| Reagent type (species) or resource | Designation | Source or reference | Identifiers | Additional Information |
|---|---|---|---|---|
| Antibody | donkey anti-rabbit Alexa 488 | Thermo-Fisher Scientific | RRID:AB_141708 | |
| Antibody | anti-rabbit TrkB H-181 | Santa Cruz | RRID:AB_2155274 | |
| Commercial assay or kit | Ovation RNA-Seq V2 kit | Nugen | Cat#: 7102 | |
| Commercial assay or kit | Ovation SoLo RNA-seq System Mouse | Nugen | Cat#: 0502–32 | |
| Commercial assay or kit | KAPA Library Quantification Kit | KAPA Biosystems | Cat#: KR0405 | |
| Commercial assay or kit | Miseq Reagent Kit v3 | Illumina | Cat#: MS-102–3001 | |
| Commercial assay or kit | Ribogreen RNA assay kit | Invitrogen | Cat#: R11490 | |
| Commercial assay or kit | RNAscope Fluorescent Multiplex V1 | Advanced Cell Diagnostics | Cat#: 320850 | |
| Chemical compound, drug | A/G magnetic beads | Pierce | Cat#: 88803 | |
| Chemical compound, drug | 10% NBF | Sigma | Cat#: HT501128 | |
| Chemical compound, drug | Fluoromount G | Southern Biotechnology | Cat#: 0100–01 | |
| Sequence-based reagent | Taqman probe *Oxt* | Life Technologies | Cat#: Mm01329577_g1 | |
| Sequence-based reagent | Taqman probe *Oxtr* | Life Technologies | Cat#: Mm01182684_m1 | |
| Sequence-based reagent | Taqman probe *Gapdh* | Life Technologies | Cat#: 4352932E | |
| Sequence-based reagent | Taqman probe *Myo5a* | Life Technologies | Cat#: Mm00487823_m1 | |
| Sequence-based reagent | Taqman probe *Ank2* | Life Technologies | Cat#: Mm00618325_m1 | |
| Sequence-based reagent | Taqman probe *Peg3* | Life Technologies | Cat#: Mm01337379_m1 | |
| Sequence-based reagent | Taqman probe *Kmt2a* | Life Technologies | Cat#: Mm01179235_m1 | |
| Sequence-based reagent | Taqman probe *Agrp* | Life Technologies | Cat#: Mm00475829_g1 | |
| Sequence-based reagent | Taqman probe *Cartpt* | Life Technologies | Cat#: Mm04210469_m1 | |
| Sequence-based reagent | RNAscope probe *Oxt* | Advanced Cell Diagnostics | Cat#: 493171 | |
| Sequence-based reagent | RNAscope probe *Ntrk2* | Advanced Cell Diagnostics | Cat#: 423611 | |
| Sequence-based reagent | RNAscope probe *Gabra2* | Advanced Cell Diagnostics | Cat#: 435011 | |
| Sequence-based reagent | RNAscope probe *Cckar* | Advanced Cell Diagnostics | Cat#: 313751 | |
| Sequence-based reagent | RNAscope probe *Ntrk2* (Exon S) | Advanced Cell Diagnostics | Cat#: 539481 | |
| Sequence-based reagent | RNAscope probe *Kmt2a* | Advanced Cell Diagnostics | Cat#: 408951 | |

*Continued on next page*

*Continued*

| Reagent type (species) or resource | Designation | Source or reference | Identifiers | Additional Information |
|---|---|---|---|---|
| Sequence-based reagent | RNAscope probe *Peg3* | Advanced Cell Diagnostics | Cat#: 492581 | |
| Sequence-based reagent | RNAscope probe *Ank2* | Advanced Cell Diagnostics | Cat#: 413221 | |

## Animals

Mice with disruption of BDNF production from promoters I, II, IV, or VI (Bdnf-e1, -e2, -e4, -e6 mice -/-, respectively) were generated as previously described (*Maynard et al., 2016*) and backcrossed to C57BL/6J > 12x. Briefly, an enhanced green fluorescent protein (eGFP)-STOP cassette was inserted upstream of the respective 5'UTR splice donor site of the targeted exon. For example, in Bdnf-e1, -e2, -e4, and –e6 -/- mice, transcription is initiated from promoter I, II, IV, or VI, producing a 5'UTR-eGFP-STOP-*Bdnf* IX transcript, which leads to GFP production in lieu of BDNF from the targeted promoter (*Figure 1b*). Sexually mature female WT and *Bdnf* -/- mice were used for all behavioral experiments.

We selectively ablated TrkB in OXT-expressing cells by crossing mice driving Cre-recombinase under control of the endogenous *Oxt* promoter, (Oxt^tm1.1(cre)Dolsn; referenced in text as Oxt$^{Cre}$, stock #024234, Jackson Labs, Bar Harbor, ME [*Wu et al., 2012*]), to mice carrying a *loxP*-flanked TrkB allele[9] (strain fB/fB, referenced in text as TrkB$^{flox/flox}$ [*Baydyuk et al., 2011*; *Grishanin et al., 2008*]). Oxt$^{Cre}$ mice were received from Jackson Labs on a mixed C57BL/6 × 129S background. TrkB$^{flox/flox}$ mice were maintained on a C57BL/6J background. For pup retrieval experiments (*Figure 3*), TrkB$^{flox/flox}$ mice (control group) and Oxt$^{Cre}$;TrkB$^{flox/flox}$ mice (experimental group) were littermates derived from the following cross: Oxt$^{Cre}$;TrkB$^{flox/flox}$ male vs.TrkB$^{flox/flox}$ females. In *Figure 3—figure supplement 3*, WT and Oxt$^{Cre}$ littermate females were generated by crossing an Oxt$^{Cre}$ male to C57BL/6J females.

For RiboTag experiments in wild-type OXT neurons, Oxt$^{Cre}$ mice were crossed to the RiboTag mouse (B6N.129-Rpl22^tm1.1Psam/J; referenced in text as Rpl22$^{HA}$, stock #011029, Jackson Labs [*Sanz et al., 2009*]), which expresses a hemagglutinin (HA) tag on the ribosomal protein RPL22 (RPL22$^{HA}$) under control of Cre-recombinase (*Figure 4a*). For RiboTag experiments in interneurons, Rpl22$^{HA}$ mice were also crossed to mice expressing Cre-recombinase under control of the endogenous cortistatin promoter (Cort^tm1(cre)Zjh/J; referenced in text as Cort$^{cre}$, stock# 010910, Jackson Labs [*Taniguchi et al., 2011*]). For RiboTag experiments in Bdnf-e1 -/- and controls (*Figure 5a*), AAV1-DIO-RPL22$^{HA}$-GFP (generous gift of McKnight lab (*Sanz et al., 2015*); packaged by Penn Vector Core, University of Pennsylvania, Philadelphia, PA), was virally injected into Bdnf-e1; Oxt$^{Cre}$ mice that were triple crossed to a mouse expressing tdTomato under control of Cre-recombinase (B6.Cg-Gt(ROSA)26Sor^tm14(CAG-tdTomato)Hze/J, referred to as tdTom, stock # 007914, Jackson Labs).

Adult female mice were housed in a temperature-controlled environment with a 12:12 light/dark cycle and *ad libitum* access to food and water. Prior to experimentation, mice were grouped housed based on genotype. For experiments in postpartum mothers, mice were impregnated by CD1 males (21 – 24 g; Envigo, Frederick, MD) and then isolated into their own cages 3 – 4 days prior to parturition. For experiments in virgin females, experimental animals were isolated into individual cages 24 hr prior to pup retrieval testing. Adult CD1 females were used for generating foreign pups (P0 to P1). All experimental animal procedures were approved by the Sobran Biosciences Institutional Animal Care and Use Committee.

## Immunohistochemistry

For verification of RPL22$^{HA}$ expression in Bdnf-e1; Oxt$^{Cre}$; tdTom mice, a representative animal was anesthetized with isoflurane ~3 weeks following intracranial infusion of AAV-DIO-RPL22$^{HA}$-GFP and transcardially perfused with 4% paraformaldehyde (PFA) in phosphate buffered saline (PBS) pH 7.4. The brain was removed, post-fixed overnight in PFA at 4°C, cryoprotected in 30% sucrose, and cut at 50 μm on a microtome (Leica, Wetzlar, Germany) equipped with a freezing stage (Physitemp, Clifton, NJ). Free-floating slices were permeabilized with 0.3% Triton/PBS and blocked in 3% Normal Goat Serum (NGS)/0.3% Triton/PBS for 1 hr. Sections were incubated in anti-mouse HA (1:000,

MMS-101R, Covance, Princeton, NJ) overnight at 4°C. The following day, sections were rinsed 3 × 10 min in PBS and incubated in goat anti-mouse Alexa 647 (Cat # A-21235, Thermo-Fisher Scientific, 1:750) for 1 hr at room temperature. Slices were washed 3 × 10 min with PBS, incubated in DAPI at 1:5000, and mounted using Fluoromount G (Cat # 0100–01 Southern Biotechnology, Birmingham, Alabama).

For OXT and cFOS quantification, adult WT and Bdnf-e1 -/- postpartum mice were sacrificed 2 hr following pup retrieval testing (1 day after parturition) and transcardially perfused with 4% PFA. Brains were removed, post-fixed overnight in PFA at 4°C, cryoprotected in 30% sucrose, and cut at 50 μm on a microtome equipped with a freezing stage. For fluorescence co-labeling experiments, free-floating sections were permeabilized with 0.5% Tween/PBS (PBST) for 30 min and blocked with 2.5% Normal Donkey Serum (NDS)/2.5% NGS/0.5% Tween in PBS for ~5 hr. Sections were then incubated with anti-mouse OXT (1:100; generous gift of Dr. Harold Gainer; NIH, Bethesda [*Ben-Barak et al., 1985*]) and anti-rabbit cFOS (1:1000, Cat # ABE457 Millipore, Massachusetts, USA) in block overnight at 4°C. Sections were rinsed 3 × 20 min with PBST, incubated in donkey anti-rabbit Alexa 488 (Cat # A-21206 ThermoFisher Scientific, 1:1000) and goat anti-mouse Alexa 647 (1:750) for 2 hr at room temperature, and then rinsed again for 4 × 15 min. Before coverslipping with Fluoromount G sections were incubated with DAPI at 1:5000 for 20 min to label nuclei.

For OXT and TrkB co-localization experiments (*Figure 3—figure supplement 1*), adult Oxt$^{Cre}$; TrkB$^{flox/flox}$, Oxt$^{Cre}$, and TrkB$^{flox/flox}$ mice were transcardially perfused with 4% PFA. Brains were removed, post-fixed overnight in PFA at 4°C, cryoprotected in 30% sucrose, and cut at 50 μm. Free-floating sections were washed 3 × 5 min in PBS and incubated in 50 mM ammonium chloride (Sigma) for 1 hr at room temperature with shaking. Sections were washed 5 min with PBS and blocked in 10% Normal Goat Serum(NGS)/0.1%Triton in PBS for 30 min at room temperature. After blocking, sections were incubated overnight in anti-mouse OXT (1:500) and anti-rabbit TrkB H-181 (1:300; sc-8316; Santa Cruz, Texas, USA) in 2% NGS/PBS at 4°C with shaking. Following overnight incubation, sections were rinsed 3 × 5 min with PBS, incubated in goat anti-rabbit Alexa 555 (1:800) and goat anti-mouse Alexa 488 (1:800) for 1 hr at room temperature in 2% NGS/PBS, and then rinsed again for 3 × 5 min. Before coverslipping with Fluoromount G, sections were incubated with DAPI at 1:5000 for 20 min to label nuclei.

## Image acquisition and analysis

WT and Bdnf-e1 -/- paraventricular nuclei (PVN, n = 6 – 9 images per animal, n = 3 animals per genotype, n = 48 total) were tile imaged in z-series at 20x magnification using a Zeiss LSM 700 microscope (Carl Zeiss, Oberkochen, Germany). Images were stitched in x,y using Zen software (Zeiss) and analyzed using custom MATLAB and R scripts. PVN regions were segmented to include the bulk of OXT-expressing neurons excluding the third ventricle and ventricular zone cells. Adaptive 3D segmentation was performed on image stacks using the CellSegm MATLAB toolbox (*Hodneland et al., 2013*) of the PVN region (filter radius = 11 px, adaptive threshold = 1e-5). Individual nuclei were further split and separated using the DAPI channel and the 3D watershed function in MATLAB as previously described (*Ram et al., 2012*) to characterize OXT and cFOS co-localization. Statistical analyses of extracted co-localization and shape characteristics were performed in R using mixture models to account for repeated measurements of the PVN region from the same animal. All processing and analysis scripts are available on GitHub (https://github.com/LieberInstitute/cFOS_OXT_Image_Analysis [*Jaffe, 2018a*; copy archived at https://github.com/elifesciences-publications/cFOS_OXT_Image_Analysis]) and data are available upon request from the authors.

## RNA extraction and qPCR

WT and Bdnf-e1 -/- postpartum mice were cervically dislocated and hypothalami were collected on ice. RNA was extracted using TRIzol (Life Technologies, Carlsbad, CA), purified using RNeasy mini-columns (Qiagen, Valencia, CA), and quantified using a Nanodrop spectrophotometer (Agilent Technologies, Savages, MD). RNA was then normalized in concentration and reversed transcribed using Superscript III (Life Technologies). Quantitative PCR (qPCR) for oxytocin (*Oxt*) and the oxytocin receptor (*OxtR*) was performed using a Realplex thermocycler (Eppendorf, Hamburg, Germany) with GEMM mastermix (Life Technologies) and 40 ng of synthesized cDNA. Individual mRNA levels were normalized for each well to *Gapdh* mRNA levels. For validation of genes enriched in OXT neurons

(*Figure 4—figure supplement 1*), cDNA was synthesized using the Ovation RNA Amplification System V2 kit (described below) and qPCR was performed as above. Taqman probes were commercially available from Life Technologies (Mm01329577_g1, Mm01182684_m1, 4352932E, Mm00487823_m1, Mm00618325_m1, Mm01337379_m1, Mm01179235_m1, Mm00475829_g1, Mm04210469_m1).

## RNAscope in situ hybridization

An adult WT virgin female was cervically dislocated and the brain was removed from the skull, flash frozen in isopentane, and stored at −80°C. Brain tissue was equilibrated to −20°C in a cryostat (Leica, Wetzlar, Germany) and serial sections of paraventricular nucleus (PVN) were collected at 16 µm. Sections were stored at −80°C until completion of the RNAscope assay.

We performed in situ hybridization with RNAscope technology utilizing the RNAscope Fluorescent Multiplex Kit V1 (Cat # 320850 Advanced Cell Diagnostics [ACD], Hayward, California) according to manufacturer's instructions. Briefly, tissue sections were fixed with a 10% neutral buffered formalin solution (Cat # HT501128 Sigma-Aldrich, St. Louis, Missouri) for 20 min at room temperature and pretreated with protease IV for 20 min. Sections were incubated with a custom-designed Channel 1 *Oxt* probe (Cat # 493171, ACD) and a commercially available *Ntrk2* (TrkB) probe (Cat # 423611-C2 Advanced Cell Diagnostics, Hayward, California). Probes were fluorescently labeled with orange (excitation 550 nm), green (excitation 488 nm), or far red (excitation 647) fluorophores using the Amp 4 Alt B-FL. Confocal images were acquired in z-series at 40x magnification using a Zeiss 700LSM confocal microscope. For co-localization of *Gabra2* and *Cckar* transcripts with *Oxt* transcripts (*Figure 5*), brains were harvested from representative adult virgin Bdnf-e1 +/+; Oxt$^{Cre}$; tdTom and Bdnf-e1 -/-; Oxt$^{Cre}$; tdTom mice and subjected to the protocol described above. Sections were incubated with commercially available probes for *Gabra2* and *Cckar* (Cat # 435011 and 313751, ACD). To evaluate loss of TrkB in OXT neurons of Oxt$^{Cre}$;TrkB$^{flox/flox}$ mice (*Figure 3—figure supplement 2*), representative virgin female Oxt$^{Cre}$, TrkB$^{flox/flox}$, and Oxt$^{Cre}$;TrkB$^{flox/flox}$ mice were cervically dislocated and brains were removed from the skull and flash frozen in isopentane. RNAscope was performed as described above using custom designed probes against *Oxt* and *Ntrk2* (Cat # 493171 and 539481, ACD). The *Ntrk2* probe was designed to target 335–756 bp of NM_001025074.2, which is the floxed region in TrkB$^{flox/flox}$ mice (*Baydyuk et al., 2011*), and therefore should not bind in OXT-expressing cells in Oxt$^{Cre}$;TrkB$^{flox/flox}$ mice. To independently validate selected genes enriched in OXT neurons (*Figure 4—figure supplement 1*), RNAscope for *Oxt*, *Kmt2a*, *Peg3*, and *Ank2* (Cat # 493171, 408951, 492581, and 413221, ACD) was performed in adult WT virgin female mice.

## Postpartum pup retrieval

Adult female mice (7 – 8 weeks old) were impregnated by CD1 males and single-housed with a compact cotton nestlet 4 – 5 days before parturition. Upon parturition, date of birth, pup number, and nest quality was recorded without disturbing cages. Pup survival was monitored until P3. Postpartum pup retrieval was performed at P1 as previously described (*Carlier et al., 1982*). Briefly, dams were isolated from the homecage for ~1 min while pups were carefully moved 20 cm from the nest. Dams were returned to the homecage and behavior was recorded for 15 min using the CaptureStar software (Clever Systems, Reston, VA). Maternal behavior was scored using both quantitative and qualitative scales with experimenter blinded to genotype. Quantitative measurements were adapted from *Carlier et al. (1982)* and included latency to first contact, latency to first retrieval, duration of carrying first pup, duration of nesting with first pup, number of move-aways from pups without retrieving, and duration of nesting with pups after completing retrieval (*Carlier et al., 1982*).

Qualitative measurements included indices for maternal behavior, abnormal behavior, pup retrieval, and global parental type. For maternal behavior index (0 to 5), dams received one point for each observed affiliative behavior, including pup contact, pup carrying, completed pup retrieval (i.e. bringing all pups back to nest), nurturing (licking, crouching, or nesting with pups for at least 1 min), and covering pups with nesting materials. For abnormal behavior index (0 to 5), dams received one point for each observed harmful or disorganized behavior, including kicking pups, biting pups, failing to retrieve all pups, repeatedly carrying pups (i.e. retrieving a pup to the nest, but then moving it out again), and misplaced retrieving (i.e. bringing pups to a random location). For retrieval index

(1 to 4), dams were ranked as following: 1-no retrieval (ignore pups), 2-partial pup retrieval (retrieve some, but not all pups), 3-complete pup retrieval (retrieve all pups, but may subsequently scatter or fail to nest with pups), 4-retrieval all pups and nurture (nest for at least 2 min with pups). For global maternal type, dams were ranked as the following: 1-parenting (retrieve all pups to nest or delivered nest to pups; nurtured or nested for at least 1 min), 2-disorganized parenting (retrieve all pups to nest, but scatter, kick, or repeatedly pick-up), 3-partial parenting (incomplete retrieval, but no scattering or repeated pick ups), 4-disorganized partial parenting (incomplete retrieval with scattering, kicking, or repeated pick ups), 5-non-parenting (no retrieval; ignore pups), 6-attack (aggressive biting).

## Virgin pup retrieval

Virgin female mice were single-housed with a compact nestlet ~24 hr before testing. During testing, three foreign pups (P0-P1) obtained from time-pregnant CD1 females (Envigo) were placed in the homecage opposite to the majority of nesting material. Behavior was recorded for 15 min using CaptureStar software, unless a pup was attacked, in which case the experiment was terminated immediately. Videos were scored blinded to genotype using the same scales described above for postpartum pup retrieval, including maternal behavior, abnormal behavior, and retrieval indices. For virgins, global parental type was modified to the following criteria based on differences in styles observed between postpartum and virgin females: 1-parenting (retrieve all pups to nest or delivered nest to pups; nurtured or nested for at least 1 min), 2-partial parenting (retrieve some but not all pups to nest), 3-irregular parenting (no retrieval but nurturing such as crouching and licking), 4-non-parenting (no retrieval; ignore pups), 5-attack (aggressive biting).

## Estrous testing

Virgin WT and Bdnf-e1 -/- female mice aged 7 – 8 weeks received vaginal cytology smears for 10 consecutive days at 2pm to determine estrous cycle patterns. Briefly, 100 µl of sterile saline was used to collect vaginal cells in a graduated pipette tip. Samples were placed in a cell culture well and assayed under a light microscope. Cell density and cytoarchitecture were used to classify estrous stage: proestrous (P), estrous (E), or diestrous/metestrous (D/M) as previously described (*McLean et al., 2012*).

## Sexual receptivity/mating experiments

WT and Bdnf-e1 -/- virgin females accustomed to handling associated with vaginal cytology smears were subjected to mating experiments. Females determined to be in estrous by characterization of vaginal cells were placed in a cage with a CD1 male mouse at the start of the dark cycle (6pm). Using infrared lights and CaptureStar software (CleverSystems), male-female interactions were video-recorded for 30 min. Duration of time spent exploring the cage or being chased/cornered by males was hand-scored with experimenter blinded to genotype. Number of female rejections (i.e. kicking, biting, lordosis failure) was also scored.

## Behavioral statistics

Statistical analysis was conducted using GraphPad Prism Software (La Jolla, CA) or R for Cox regression, Mann-Whitney-Wilcoxon rank sum test, multinomial regression, or log rank survival tests where appropriate. Comparisons between two groups were performed using unpaired Student's t-test or Mann-Whitney tests where appropriate. Comparisons between three or more groups were performed using ANOVA with Bonferroni posthoc tests or Kruskal-Wallis with Dunn's multiple comparisons. Data are presented as mean ± SEM and statistical significance was set at $*p < 0.05$, $**p < 0.01$, and $p < 0.001$, and $\#p < 0.0001$.

## RiboTag and RNA-sequencing

To identify genes enriched in OXT neurons, hypothalami of $Oxt^{Cre}$; $Rpl22^{HA}$ mice were collected and flash frozen in isopentane. For each sample (n = 3 Input and n = 3 IP), five hypothalami were pooled and homogenized according to previously described protocols (*Sanz et al., 2013*). Thirty microliters of total homogenate were flash frozen and reserved for 'Input' samples. Ribosome-mRNA complexes ('IP' samples) were affinity purified using a mouse monoclonal HA antibody (MMS-101R,

Covance, Princeton, NJ) and A/G magnetic beads (88803 Pierce). RNA from Input and IP samples was purified using RNeasy microcolumns (Qiagen, Valencia, CA) and quantified using the Ribogreen RNA assay kit (R11490 Invitrogen). The Ovation RNA Amplification System V2 kit (7102 Nugen, San Carlos, CA) was used to amplify cDNA from 7 to 8 ng of RNA according to manufacturer's instructions. cDNA was used for qPCR validation for *Oxt* enrichment in IP versus Input samples and to generate sequencing libraries with the Nextera DNA Library Preparation kit. Libraries were sequenced on a Hiseq 3000 according to manufacturer's instructions. To replicate these results and establish sequencing workflows for lower input samples, we also generated libraries from these exact samples using the Ovation SoLo RNA-seq System Mouse (0502–32 Nugen, San Carlos, CA) according to manufacturer's instructions from 7 to 8 ng of starting material. Library concentration was quantified using the KAPA Library Quantification Kit (KR0405, KAPA Biosystems, Wilmington, MA). Libraries were sequenced using the MiSeq Reagent Kit v3 (MS-102–3001 Illumina, San Diego, CA) and Nugen Custom SoLo primer.

To compare differentially expressed genes in OXT neurons between control and Bdnf-e1-/- females, an adeno-associated viral vector, AAV1-DIO-RPL22$^{HA}$-GFP was virally injected into the PVN of Bdnf-e1 +/+; Oxt$^{Cre}$; tdTom or Bdnf-e1 -/-; Oxt$^{Cre}$; tdTom virgin females (*Figure 5a*). A viral approach was used due to non-Mendelian generation of expected genotypes for Bdnf-e1; Oxt$^{Cre}$; Rpl22$^{HA}$ triple crosses. Briefly, mice were anesthetized with isoflurane and mounted on a stereotaxic frame (Kopf, Tujunga, CA). Virus was bilaterally injected into the PVN using a Hamilton syringe at an injection rate of 100 nL/min. PVN coordinates were anteroposterior (AP): −0.7 mm, mediolateral (ML):±0.3 mm, and dorsoventral (DV): −4.75 mm. Injection volumes were 500 nL. Three weeks following surgery, mice were sacrificed and hypothalami were dissected and homogenized individually and processed as described above. IP sequencing libraries (n = 3 control and n = 3 Bdnf-e1 -/-) were prepared using the Ovation SoLo RNA-seq System Mouse according to manufacturer's instructions from less than 1 ng of starting material. Library concentration was quantified using the KAPA Library Quantification Kit (KR0405, KAPA Biosystems, Wilmington, MA). Libraries were sequenced using the MiSeq Reagent Kit v3 (MS-102 – 3001 Illumina, San Diego, CA) and Nugen Custom SoLo primer.

## RNA-seq data processing and analyses

RNA-seq reads from all experiments were aligned and quantified using a common processing pipeline. Reads were aligned to the mm10 genome using the HISAT2 splice-aware aligner (*Kim et al., 2015*) and alignments overlapping genes were counted using featureCounts version 1.5.0-p3 (*Liao et al., 2014*) relative to Gencode version M11 (118,925 transcripts across 48,709 genes, March 2016). Read counting mode was used for single end read libraries (SoLo) and fragment counting mode was used for paired end reads (Ovation, Clonetech). We analyzed 22,472 genes with reads per kilobase per million counted/assigned (RPKM normalizing to total number of gene counts not mapped reads)>0.1 in the RiboTag IP versus input analyses and 22,071 genes with RPKM >0.1 in the control vs. Bdnf-e1 analysis. Differential expression analyses were performed on gene counts using the voom approach (*Law et al., 2014*) in the limma R/Bioconductor package (*Ritchie et al., 2015*) using weighted trimmed means normalization (TMM) factors. Differential expression analyses further adjusted for the gene assignment rate/exonic mapping rate, calculated from the output of feature-Counts, which reflects the proportion of aligned reads/fragments assigned to genes during counting. Here the total proportion of exonic reads, which typically explains a large proportion of gene count variance (*Jaffe et al., 2017*) was higher (but not significantly) in IP compared to Input RNA fractions (73.2% versus 66.0%, $t_{df=4}$=2.16, p=0.10). Genes identified as consistently associated with immunoprecipitation were based on the following criteria: FDR < 1% in Oxt IP vs input analyses, and directionally consistent and marginally significant (at p<0.01) differential expression in both Cort IP vs input and Ntsr1 IP vs input. For control vs. Bdnf-e1 experiments, differential expression modeling for genotype effects further adjusted for the log$_2$ expression of *Oxt* to adjust for viral transfection differences. Multiple testing correction was performed using the Benjamini-Hochberg approach to control for the false discovery rate (FDR)(*Kasen et al., 1990*). Gene set enrichment analyses were performed using the subset of genes with known Entrez gene IDs using the clusterProfiler R Bioconductor package (*Yu et al., 2012*). All RNA-seq analysis code is available on the GitHub repository https://github.com/LieberInstitute/oxt_trap_seq (*Jaffe, 2018b*; copy archived at https://github.com/elifesciences-publications/oxt_trap_seq).

## Acknowledgements

We thank Dr. Harold Gainer (NIH, Bethesda) for generously providing the OXT antibody. We thank Dr. Stanley McKnight (University of Washington, Seattle) for the generous gift of AAV-DIO-Ribotag vector. We acknowledge the Penn Vector Core (University of Pennsylvania, Philadelphia) for viral packaging. We thank Dr. Daniel Weinberger for comments on the manuscript. Funding for these studies was provided by the Lieber Institute for Brain Development and the National Institute for Mental Health (T32MH01533037 to KRM and RO1MH105592 to KM).

## Additional information

### Funding

| Funder | Grant reference number | Author |
| --- | --- | --- |
| National Institute of Mental Health | T32MH01533037 | Kristen R Maynard |
| Lieber Institute for Brain Development | | Keri Martinowich |
| National Institute of Mental Health | RO1MH105592 | Keri Martinowich |

The funders had no role in study design, data collection and interpretation, or the decision to submit the work for publication.

### Author contributions

Kristen R Maynard, John W Hobbs, Conceptualization, Data curation, Formal analysis, Supervision, Funding acquisition, Validation, Investigation, Visualization, Methodology, Writing—original draft, Project administration, Writing—review and editing; BaDoi N Phan, Amolika Gupta, Data curation, Software, Formal analysis, Methodology, Writing—original draft; Sumita Rajpurohit, Courtney Williams, Anandita Rajpurohit, Data curation, Formal analysis; Joo Heon Shin, Data curation, Formal analysis, Supervision, Methodology; Andrew E Jaffe, Conceptualization, Data curation, Software, Formal analysis, Supervision, Methodology, Writing—original draft, Writing—review and editing; Keri Martinowich, Conceptualization, Software, Formal analysis, Supervision, Funding acquisition, Investigation, Methodology, Writing—original draft, Project administration, Writing—review and editing

### Author ORCIDs

Kristen R Maynard (iD) http://orcid.org/0000-0003-0031-8468
BaDoi N Phan (iD) http://orcid.org/0000-0001-6331-5980
Keri Martinowich (iD) http://orcid.org/0000-0002-5237-0789

### Ethics

Animal experimentation: All experimental animal procedures were approved by the Sobran Biosciences Institutional Animal Care and Use Committee (IACUC) under protocol number LIE-004-2015.

### Decision letter and Author response

Decision letter https://doi.org/10.7554/eLife.33676.022
Author response https://doi.org/10.7554/eLife.33676.023

## Additional files

### Supplementary files

• Supplementary file 1. Table Legends (Excel tables attached as separate tabs in a single file). Table S1. Differential gene expression analysis of OXT Input vs. IP samples for all expressed genes. Table S2. Comparison of significant OXT neuron gene enrichment using Nugen Ovation RNA-seq System

V2 kit vs. Nugen Ovation SoLo RNA-seq system. Table also includes comparison of differential gene expression between OXT neurons, Cort inhibitory interneurons, and Ntsr1 layer VI corticothalamic neurons versus their respective input RNA samples, as well as an overall OXT-IP versus pooled Cort-IP and Ntsr1-IP samples to assess specificity for these genes that were enriched in OXT neurons. Table S3. Gene Ontology analysis in OXT neurons. Enriched and depleted GO terms from differential expression analysis of OXT Input vs. IP. Table S4. Differential gene expression analysis of OXT-IP vs. Cort-IP and Ntsr1-IP samples for all expressed genes. Table S5. Gene ontology enrichment analysis for genes differentially expressed in OXT-IP vs. Cort-IP and Ntsr1-IP samples Table S6. OXT enriched mouse model genes in SFARI. List of genes enriched in OXT neurons implicated in genetic animal models of autism spectrum disorder (ASD) by the Simons Foundation Autism Research Initiative (SFARI). Table S7. OXT enriched human genes in SFARI. List of genes enriched in OXT neurons implicated in ASD by the SFARI Human Gene Module using the subset of mouse-expressed genes with human homologs. Table S8. Differential gene expression analysis of control IP vs. Bdnf-e1 -/- IP samples. Table S9. Gene Ontology analysis in OXT neurons with disruption of BDNF signaling. GO terms enriched from differential expression analysis of control IP vs. Bdnf-e1 -/- IP samples. Table S10. BDNF-dependent OXT genes in SFARI. List of genes perturbed in OXT neurons following disruption of BDNF signaling that are implicated in autism spectrum disorder (ASD) by the Simons Foundation Autism Research Initiative (SFARI).

DOI: https://doi.org/10.7554/eLife.33676.015

• Transparent reporting form
DOI: https://doi.org/10.7554/eLife.33676.016

## Data availability

All RNA-seq analysis code and source data are available on the GitHub repository: https://github.com/LieberInstitute/oxt_trap_seq/ (copy archived at https://github.com/elifesciences-publications/oxt_trap_seq). Raw sequencing reads are available on SRA under accession code SRP157978. All processing and analysis scripts for immunohistochemistry co-localization studies are available on GitHub: https://github.com/LieberInstitute/cFOS_OXT_Image_Analysis (copy archived at https://github.com/elifesciences-publications/cFOS_OXT_Image_Analysis).

The following dataset was generated:

| Author(s) | Year | Dataset title | Dataset URL | Database, license, and accessibility information |
|---|---|---|---|---|
| Maynard KR, Hobbs JW, Phan BN, Gupta A, Rajpurohit S, Williams C, Rajpurohit N, Shin J, Jaffe AE, Martinowich K | 2018 | BDNF-TrkB signaling in oxytocin neurons contributes to maternal behavior | https://www.ncbi.nlm.nih.gov/Traces/study/?acc=SRP157978 | Raw sequencing reads are available on SRA under accession code SRP157978 |

The following previously published dataset was used:

| Author(s) | Year | Dataset title | Dataset URL | Database, license, and accessibility information |
|---|---|---|---|---|
| Nectow AR, Moya MV, Ekstrand MI, Mousa A, McGuire KL, Sferrazza CE, Field BC, Rabinowitz GS, Sawicka K, Liang Y, Friedman JM, Heintz N, Schmidt EF | 2017 | Rapid molecular profiling of defined cell types using viral TRAP | http://www.ncbi.nlm.nih.gov/geo/query/acc.cgi?acc=GSE89737 | Publicly available at the NCBI Gene Expression Omnibus (accession no: GSE89737). |

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
