## [Decision Letter]

Thank you for submitting your article "BDNF-TrkB signaling in oxytocin neurons contributes to maternal behavior" for consideration by *eLife*. Your article has been reviewed by three peer reviewers, and the evaluation has been overseen by a Reviewing Editor and a Senior Editor. The following individual involved in review of your submission has agreed to reveal his identity: Eero Castrén (Reviewer #3).

The reviewers have discussed the reviews with one another and the Reviewing Editor has drafted this decision to help you prepare a revised submission.

Summary:

In this study, the authors report that deletion of BDNF from exons I, and to a lesser extent deletion of exon 2, disrupt maternal care and sexual receptivity. Loss of TrkB in oxytocin (Oxt) neurons partially recapitulates the phenotype, suggesting a contribution of BDNF-TrkB signaling in OXT neurons to maternal care behaviors. The authors use state of the art generation sequencing of OXT neurons and its changes in BDNF exon 1 knockout mice and report alterations in structural and functional plasticity pathways. While all three reviewers expressed interest in the study, there were comments that should be addressed.

Essential revisions:

1) A shared concern was linking the overall phenotype to Oxt neurons.

The phenotype of the Oxt specific TrkB knockouts is relatively modest and as the authors note is only a partial recapitulation of the phenotype in the BDNF exon 1 knockout mice. It is important for the authors to rule out the following possibilities regarding the TrkB knockouts:

A) Knockout was not complete. The authors should show loss of TrkB in Cre expressing or Oxt expressing neurons.

B) Are the TrkB mice on a different genetic background than *Oxt* and *Bdnf* mutants? Given the difference in the WT behavior between Figure 3H and the WT behavior in 1P the authors are asked to clarify this point.

C) The Oxt-Cre alone has a phenotype. Since these are knock-in mice, they might be hypomorphs for Oxt protein production, which could impact maternal behavior.

2) There was also some concern regarding the profiling of the Oxt neurons and the request to further validate this data with more independent methods (i.e. ISH for key transcripts that are predicted to be enriched by the RNAseq data). Related to this data set, the authors highlight commonalities of Oxt neurons to other neuronal populations but it would be important using the same data set to highlight the differences between OXT neurons and these neuronal populations.

3) Although not required, the authors may want to examine whether Oxt levels are reduced in the TrkB knockout mice and whether pup retrieval stimulates Fos normally in the Oxt neurons to strengthen the overall link to BDNF signaling in Oxt neurons to the reported behavior.

---

## [Author Response]

Essential revisions:1) A shared concern was linking the overall phenotype to Oxt neurons.The phenotype of the Oxt specific TrkB knockouts is relatively modest and as the authors note is only a partial recapitulation of the phenotype in the BDNF exon 1 knockout mice. It is important for the authors to rule out the following possibilities regarding the TrkB knockouts:A) Knockout was not complete. The authors should show loss of TrkB in Cre expressing or Oxt expressing neurons.

We agree with the reviewers that incomplete knockout of TrkB in OXT neurons could contribute to a milder phenotype. To address this issue, we performed immunohistochemistry for OXT and TrkB in Oxt*^Cre^*, TrkB*^flox/flox^*, and Oxt*^Cre^*; TrkB*^flox/flox^* mice using the highly specific OXT antibody (Gainer lab) and an antibody selective for TrkB (TrkB H181; Santa Cruz). We have included this new data in the manuscript as Figure 3—figure supplement 1. However, one challenge with this approach is that TrkB is expressed at both the pre- and postsynapse making it difficult to clearly identify and quantify the presence or absence of TrkB signal belonging to oxytocin cell bodies across genotypes. To address this challenge and further verify successful recombination and subsequent loss of TrkB in OXT neurons, we used single molecule fluorescent in situ hybridization to evaluate the presence or absence of *Ntrk2* transcripts in OXT neurons. Our strategy was to design a new *Ntrk2* probe that targets the floxed region in TrkB*^flox/flox^*mice. Based on this design, we expect to see a selective absence of labeling in OXT neurons in Oxt*^Cre^*; TrkB*^flox/flox^* mice. We have included this new data in the manuscript in Figure 3—figure supplement 2. In addition to these new data, we would also like to note that both previous literature and our own studies support that the Oxt*^Cre^* and TrkB*^flox/flox^* alleles are functional. For example, we demonstrate in Figure 5b that the Cre recombinase driven from the *Oxt* promoter is capable of recombining the virally delivered DIO-Ribotag vector as well as the endogenous tdTom reporter. The

TrkB*^flox/flox^*mice have also been extensively validated by other laboratories (Grishanin et al., 2008; Kotloski et al., 2010, Baydyuk et al., 2011; Zheng et al., 2011) and in our hands, crossing TrkB*^flox/flox^* mice to other Cre alleles causes severe phenotypes. For example, Cort*^Cre^*; TrkB*^flox/flox^* mice lacking TrkB in cortistatin interneurons have severe seizures resulting in death by P35 (Hill et al., under revision). Based on our new data and previous experience with these alleles, we have full confidence that recombination is occurring, and therefore TrkB is being knocked out. As TrkB is highly expressed in other cell types in the hypothalamus (Figure 3A and Figure 3— figure supplements 1 and 2), it remains likely BDNF-TrkB signaling in additional neuronal populations contributes to maternal behavior.

B) Are the TrkB mice on a different genetic background than Oxt and Bdnf mutants? Given the difference in the WT behavior between Figure 3H and the WT behavior in 1P the authors are asked to clarify this point.

We thank the reviewers for noting this difference. Our TrkB*^flox/flox^*and Bdnf-e1 colonies are maintained on a C57BL/6J background. However, the Oxt*^Cre^*mice that we obtained from Jackson labs were maintained on a mixed C57/129 background. Therefore, the mice utilized in Figure 3 are a mixed background of C57/129. As maternal care and performance on the pup retrieval test can vary by strain (Carlier et al., 1982), this likely accounts for differences in WT/control virgin behavior between Figure 3H and 1P. To limit the impact of a mixed genetic background in Figure 3, TrkB*^flox/flox^*andOxt*^Cre^*; TrkB*^flox/flox^* mice were littermates derived from the following cross: Oxt*^Cre^*; TrkB*^flox/flox^* male versus TrkB*^flox/flox^*females. We have added this information to the Materials and methods and discussed differences in penetrance as a potential contributor to a milder phenotype.

C) The Oxt-Cre alone has a phenotype. Since these are knock-in mice, they might be hypomorphs for Oxt protein production, which could impact maternal behavior.

We agree that this is an important question. In our initial submission, we used TrkB*^flox/flox^*females for our control group because 1) this fit our preferred breeding scheme above, which unfortunately does not generate Oxt*^Cre^*mice, and 2) Marlin et al., 2015 had already published pup retrieval data with Oxt*^Cre^* mice and did not discuss a potential phenotype in these animals. To rule out any concerns, we have run a separate behavioral cohort of WT and Oxt*^Cre^* littermates, which are on a mixed C57/129 background similar to the mice in Figure 3, on the virgin pup retrieval test. These animals were derived from the following cross: WT (C57BL/6J) females versus heterozygous Oxt*^Cre^* (mixed C57/129) male. We found no impairments in maternal behavior in Oxt*^Cre^*mice suggesting that the Oxt*^Cre^*allele is not driving the phenotype in Oxt*^Cre^*; TrkB*^flox/flox^* mice. The results of this experiment have been added to the manuscript in Figure 3—figure supplement 3. We would also like to highlight that our immunohistochemistry for OXT in Oxt*^Cre^*, TrkB*^flox/flox^*, and Oxt*^Cre^*; TrkB*^flox/flox^* mice (Figure 3—figure supplement 1) demonstrates that Oxt*^Cre^*mice appear to have similar levels of OXT protein compared to TrkB*^flox/flox^*, and Oxt*^Cre^*; TrkB*^flox/flox^* mice.

2) There was also some concern regarding the profiling of the Oxt neurons and the request to further validate this data with more independent methods (i.e. ISH for key transcripts that are predicted to be enriched by the RNAseq data). Related to this data set, the authors highlight commonalities of Oxt neurons to other neuronal populations but it would be important using the same data set to highlight the differences between OXT neurons and these neuronal populations.

We have further validated our RNAseq data in Figure 4 with both quantitative RT-PCR and fluorescent *in situ* hybridization using RNAscope for selected top hits, including *Ank2* and *Kmt2a*. These data have been added as the new Figure 4—figure supplement 1 and are discussed in the manuscript. We also highlight Figure 4— figure supplement 2, where the high correlation between kits replicates the entire data set as the same RNA samples were used to prepare 2 independent library preparations and sequenced on 2 different instruments. We have also provided an expanded Table S2 and new Supplementary Tables S4 and S5 in Supplementary file 1, which highlight genes and gene pathways that are different between OXT neurons and Cort and Ntsr1 neurons. Notable genes have been identified and described in the manuscript.

3) Although not required, the authors may want to examine whether Oxt levels are reduced in the TrkB knockout mice and whether pup retrieval stimulates Fos normally in the Oxt neurons to strengthen the overall link to BDNF signaling in Oxt neurons to the reported behavior.

We have made an effort to address this question in Figure 3—figure supplement 1 by using immunohistochemistry to compare OXT expression between Oxt*^Cre^*, TrkB*^flox/flox^*, and Oxt*^Cre^*; TrkB*^flox/flox^* mice. While only qualitative, it appears that the number of OXT neurons and intensity of OXT labeling is relatively normal in Oxt*^Cre^*, TrkB*^flox/flox^*mice. Unfortunately, we did not have the breeders to quickly obtain additional mature Oxt*^Cre^*; TrkB*^flox/flox^*females for pup retrieval experiments to generate the required brain tissue for cFOS staining. However, given that Bdnf-e1 females with a severe behavioral phenotype do not show changes in OXT or cFOS in adulthood, we would expect that Oxt*^Cre^*; TrkB*^flox/flox^*mice exhibiting a milder phenotype would not display these changes. While we did not have the resources to set up triple crosses to conduct Ribotag experiments in these animals, we have added this to the discussion as an important future direction.